# Optimizing Layerwise Polynomial Approximation for Efficient Private Inference on Fully Homomorphic Encryption: A Dynamic Programming Approach

## Abstract

Recent research has explored the implementation of privacy-preserving deep neural networks solely using fully homomorphic encryption. However, its practicality has been limited because of prolonged inference times. When using a pre-trained model without retraining, a major factor contributing to these prolonged inference times is the high-degree polynomial approximation of activation functions such as the ReLU function. The high-degree approximation consumes a substantial amount of homomorphic computational resources, resulting in slower inference. Unlike the previous works approximating activation functions uniformly and conservatively, this paper presents a *layerwise* degree optimization of activation functions to aggressively reduce the inference time while maintaining classification accuracy by taking into account the characteristics of each layer. Instead of the minimax approximation commonly used in state-of-the-art private inference models, we employ the weighted least squares approximation method with the input distributions of activation functions. Then, we obtain the layerwise optimized degrees for activation functions through the *dynamic programming* algorithm, considering how each layer's approximation error affects the classification accuracy of the deep neural network. Furthermore, we propose modulating the ciphertext moduli-chain layerwise to reduce the inference time. By these proposed layerwise optimization methods, we can reduce inference times for the ResNet-20 model and the ResNet-32 model by 3.44 times and 3.16 times, respectively, in comparison to the prior implementations employing uniform degree polynomials and a consistent ciphertext modulus.

## 1 Introduction

With the advancement of cloud computing, numerous data sets are being shared online, and data analysis is often carried out through data sharing via the cloud. One prominent example is Machine Learning as a Service (MLaaS), which involves performing machine learning tasks in the cloud. However, sharing data via the cloud requires significant attention to privacy concerns. Therefore, Privacy Preserving Machine Learning (PPML) is imperative to provide secure machine learning services while guaranteeing data privacy. One of the promising solutions in PPML is homomorphic encryption (HE). Clients can use this HE technique to encrypt their data, preventing the disclosure of personal information in the shared data on the cloud. By leveraging the properties of HE, servers can perform operations directly on ciphertexts without decryption. Since fully homomorphic encryption (FHE) allows unlimited operations on ciphertext, it is particularly well-suited for deep learning tasks requiring long sequential computations for many layers. (Gilad-Bachrach et al., 2016; Boemer et al., 2019; Al Badawi et al., 2020; Lee et al., 2022a; 2023a; Kim & Guyot, 2023; Lee et al., 2023b)

The main problem in recently implemented PPML models (Lee et al., 2022b;a; Kim & Guyot, 2023) on FHE is the long runtime, primarily due to the computation of the non-arithmetic activation function on FHE. While deep learning models can achieve high performance by incorporating non-arithmetic activation functions (e.g., ReLU, sigmoid, and GELU), the existing FHE schemes support only basic arithmetic operations, such as addition and multiplication. This limitation poses

a challenge when conducting inference using deep learning models within the FHE framework. To address the constraints posed by FHE, some research in PPML (Gilad-Bachrach et al., 2016; Chou et al., 2018; Al Badawi et al., 2020) replaced the ReLU function with low-degree polynomial activation functions, which can be easily computed on FHE. However, this approach necessitates retraining of the model, a process that can additionally consume significant computation resources and datasets. Furthermore, these non-standard simplified models have been shown to be effective primarily in simple datasets and have not demonstrated their efficacy in large and practical datasets.

To utilize the standard pre-trained models on FHE'ed data without any retraining, the state-of-the-art works (Lee et al., 2022a;b) have employed high-degree polynomials to approximate non-linear activation functions accurately. Lee et al. (2022a;b) successfully implemented ResNet models on FHE, utilizing a composite polynomial approximation technique for the sign function (Lee et al., 2021), which ensures minimal homomorphic operations required for precise sign function approximation. While this approach offers the advantage of obviating the need for model redesign or retraining, it does entail the use of a high-degree polynomial, resulting in significantly prolonged inference time.

In this work, we aim to aggressively reduce the inference time while utilizing pre-trained standard models without the need for retraining. The approximate polynomial in prior works (Lee et al., 2022a;b; Kim & Guyot, 2023) was determined without adequately considering the characteristics of each layer of deep neural networks, rendering these implementations less efficient. We point out that each layer exhibits two distinct characteristics closely related to the activation approximation. First, the input distribution for the activation function in each layer is close to the Gaussian distribution rather than the uniform distribution, in contrast to the previous composite approximation techniques (Lee et al., 2021), which approximates the activation function uniformly in the approximation region. Second, the impact of the *approximation* accuracy of activation functions on actual *classification* accuracy varies from layer to layer. Essentially, the approximation accuracy of the activation functions significantly impacts the classification accuracy of the neural network in particular layers. In other layers, employing a less accurate approximation of the activation function does not substantially affect classification accuracy. The previous works have yet to take into account the differing impacts at different layers and subsequently employed the identical polynomial across all layers.

In this paper, we introduce a principled polynomial approximation method for activation functions to optimize the overall inference runtime of deep neural networks on FHE, considering each layer's unique characteristics. In essence, we propose a *layerwise* approximation method for activation functions that optimizes the polynomial degree for each layer without necessitating retraining or altering the standard deep learning models in homomorphic encryption contexts. Our contributions can be summarized as follows:

- Unlike the conventional minimax approximation, we advocate a method that employs the *weighted least squares method*. This approach achieves precise polynomial approximations in high-probability regions and less precise approximations in low-probability regions. This results in high classification accuracy with lower-degree polynomials with degree 63 at maximum for the ResNet-20, compared to degree 6,075 with the previous minimax approximation.

- We present a *dynamic programming algorithm* designed to determine the optimal degree set for each layer. This algorithm efficiently obtains the optimal polynomial degrees, minimizing the overall negative impact of polynomial approximations on classification accuracy while simultaneously bounding the inference runtime of the deep neural network on encrypted data.

- By introducing a *moduli-chain management method*, we achieve additional reductions in inference runtime. This method focuses on removing unused moduli, a consequence of the variations in the optimized polynomial degrees, through the proposed dynamic programming algorithm.

- We conduct simulations of the ResNet models on encrypted data with the residual number system Cheon-Kim-Kim-Song (RNS-CKKS) scheme using the `Lattigo` library (Lattigo, 2022). Our experiments confirm that our principled polynomial approximation technique can reduce the inference runtime of the ResNet-20 and the ResNet-32 models by 3.44 times and 3.16 times, respectively, while maintaining classification accuracy (Lee et al., 2022b).

## 2 WEIGHTED LEAST SQUARES APPROXIMATION

### 2.1 MOTIVATION

Since the FHE scheme only supports arithmetic operations such as addition and multiplication, non-arithmetic functions must be approximated by polynomials to perform the deep neural network on FHE. Most previous works (Lee et al., 2022b;a; Kim & Guyot, 2023) use the minimax approximation method, which optimizes approximation based on the $\ell_\infty$ norm within a designated approximation region. This previous method has two characteristics: (i) it attempts to provide consistent approximation accuracy for all points within its designated approximation region, and (ii) its accuracy declines rapidly outside this region due to the divergence of approximate polynomials. A single input of an activation function, if it falls outside the approximation region, can induce a significant deviation caused by the approximate polynomial, potentially leading to an erroneous inference outcome. Therefore, it is required to encompass even low-probability values within the approximation region to maintain classification accuracy. This necessitates approximating a broader region than where most values lie. Given that the minimax approximation method provides consistent accuracy within its approximation region, it tends to over-approximate with excessive precision in low-probability regions. A polynomial with a lower degree, capable of precisely approximating the activation function in high-probability regions and providing a satisfactory approximation in low-probability regions, would be more efficient than those obtained using the previous minimax approximation method.

### 2.2 POLYNOMIAL APPROXIMATION BY WEIGHTED LEAST SQUARES METHOD

We propose the use of the weighted least squares method for approximating activation functions. Unlike the previous minimax approximation, the weighted least squares approach takes into account the input distribution specific to each activation function during approximation.

Consider $f(x)$ as the non-arithmetic activation function to be approximated as polynomials. Suppose that the input data for $f(x)$ in the $i$th layer follows a distribution $\phi_i(x)$. Assuming that the distribution of the test dataset matches the training dataset, then the input distribution $\phi_i(x)$ can be inferred from the training dataset. The proposed polynomial approximation method then aims to minimize the mean squared error (MSE), as formulated by:

$$\mathbb{E}_{x \sim \phi_i}[(f(x) - p_i(x))^2] = \int_\mathbb{R} \phi_i(x)(f(x) - p_i(x))^2 dx, \tag{1}$$

where the activation function $f(x)$ in the $i$th layer is approximated as the polynomial $p_i(x)$. By minimizing the MSE given in Equation 1, the reduction of approximation error $f(x) - p_i(x)$ becomes more pronounced in the high-probability regions corresponding to larger values of $\phi_i(x)$.

The runtime of polynomial evaluation on FHE is determined by the polynomial's degree. In addition, precise inference in deep neural networks requires minimizing the MSE. Thus, given a polynomial degree $d$, it is critical to select the polynomial that achieves the minimum MSE. Under general conditions, such a polynomial invariably exists, as articulated in the following theorem.

**Theorem 2.1.** *Let $f(x)$ be a function and $\phi(x)$ be an input distribution function, and suppose that $\int_\mathbb{R} \phi(x)f(x)^2 dx < \infty$. Then, for each non-negative integer $d$, there exists a polynomial $P[d;f](x)$, which minimizes the MSE over the polynomials with a degree not greater than $d$. In other words, there exists a polynomial $P[d;f](x)$ with a degree not greater than $d$ which satisfies*

$$\int_\mathbb{R} \phi(x)(f(x) - P[d;f](x))^2 dx \leq \int_\mathbb{R} \phi(x)(f(x) - q(x))^2 dx$$

*for any polynomial $q(x)$ with a degree not greater than $d$.*

Additionally, if the input distribution is normal, the approximate polynomial can be derived in closed form. The following theorem outlines how to obtain such polynomials under the normality assumptions.

**Theorem 2.2.** *If the input distribution $\phi(x)$ is normal with mean $\mu$ and variance $\sigma^2$ (i.e., $\phi(x) = \frac{1}{\sqrt{2\pi\sigma^2}}e^{-\frac{(x-\mu)^2}{2\sigma^2}}$), the approximate polynomial for the function $f(x)$*

$$P_{\mu,\sigma^2}[d;f](x) := \sum_{l=0}^{d} h_l\left(\frac{x-\mu}{\sigma}\right)\int_{\mathbb{R}} \phi(t)f(t)h_l\left(\frac{t-\mu}{\sigma}\right)dx$$

*minimizes the MSE over the polynomials with a degree not greater than $d$, where $h_l(x) = \frac{(-1)^l}{\sqrt{l!}}e^{x^2/2}\frac{d^l}{dx^l}e^{-x^2/2}$ is the polynomial with degree $l$. Also, the MSE of the polynomial $P_{\mu,\sigma^2}[d;f](x)$ is given by*

$$E_{\mu,\sigma^2}[d;f] = \int_{\mathbb{R}} \phi(x)f(x)^2dx - \sum_{i=0}^{d}\left[\int_{\mathbb{R}} \phi(x)f(x)h_l\left(\frac{x-\mu}{\sigma}\right)dx\right]^2.$$

We provide the details of the proof for Theorem 2.2 in Appendix B. We use the polynomial $P_{\mu,\sigma^2}[d;f](x)$ for practical approximate polynomial instead of the non-arithmetic activation functions, as the input distribution for each activation function usually follows the normal distribution. Also, we formulate an optimization problem to determine the most effective polynomial degrees based on the MSE $E_{\mu,\sigma^2}[d;f]$ in Section 3.

**Scaling $\sigma$.** We approximate the activation function $f(x)$ considering the distribution of its input values. However, during inference with approximate polynomials $P_{\mu,\sigma^2}[d;f](x)$, the input distribution of those polynomials may be perturbed by approximation errors. Such deviation can lead to critical errors, given the divergence of the approximation polynomials. Therefore, for inference on the test dataset, we scale $\sigma$ to $r\sigma$ for $r > 1$ to decrease the probability of divergence in the output of the polynomial $P_{\mu,(r\sigma)^2}[d;f](x)$.

### 2.3 ADJUSTMENT OF THE COEFFICIENTS' MAGNITUDES

We propose an additional method to reduce the coefficients' magitudes, ensuring stable polynomial operations for ciphertexts. Unlike in the plaintext domain, it is essential that input values fall within the interval $[-1, 1]$ to enable stable polynomial evaluation in the ciphertext domain. To direct input values to fall within the range of $[-1, 1]$, we first set the evaluation region $[B_1, B_2]$, which covers all input values of each activation function. Then, we change the scale of the approximate polynomial to align with the transition from evaluation region $[B_1, B_2]$ to $[-1, 1]$. [1]

However, this transformation makes the coefficients quite large. The coefficient of $x^l$ is proportional to $(B_2 - B_1)^l$ when the center of the region is fixed. (See Theorem D.1 in Appendix D.) To deal with this issue, we propose to scale again, from $r\sigma$ to $\lambda r\sigma$ for a constant $\lambda > 1$. When $\lambda > 1$, the coefficients of the high-degree term decrease rapidly, proportional to $\lambda^{-l}$. (See Theorem D.2 in Appendix D.) We discuss the effect of adopting $\lambda$ by simulation results in Section 5.

## 3 DYNAMIC PROGRAMMING FOR LAYERWISE DEGREE OPTIMIZATION

In this section, we propose a *layerwise* approximation method to optimize the degrees of approximate polynomials. By quantifying the impact of each layer's approximation error on classification accuracy, we formulate an optimization problem to determine the optimal degrees for each layer's polynomials. Subsequently, we devise an algorithm to solve this optimization problem via a *dynamic programming approach*, providing the optimized set of degrees with polynomial time complexity.

### 3.1 MINIMIZING THE VARIANCE OF THE LOSS FUNCTION

To assess how the approximation error in each layer affects classification accuracy, we should decide on an appropriate surrogate function closely related to classification accuracy. We propose to use the *variance of the loss* as a surrogate function in our optimization problem, given that the training process aims to minimize its loss function.

---

[1]See Appendix A for polynomial evaluation on FHE and Appendix D to check how we adapt the approximate polynomial.

Assuming that the pre-trained model under plaintext has successfully minimized the loss, substituting the activation function with the approximate polynomials for the encrypted data introduces approximation errors that alter the optimized loss. This change can have a negative impact on classification accuracy. We use a relaxation that the approximation error of each activation function is treated as a random variable, although it is actually deterministic for a given input value. This assumption enables us to quantify the relation between the classification accuracy and the approximation error of each activation function. The justification for this relaxation lies in the consideration that the approximation error can be regarded as a random variable across different input data of all layers.

Consider a pre-trained model with a loss function $\mathcal{L}$. This loss function depends on the activation nodes and can be expressed as $\mathcal{L}(\{a_{i,j}\})$, where $a_{i,j}$ denotes the $j$th activation node in the $i$th layer ($i = 1, \cdots, N_L$, $j = 1, \cdots, n_i$). When we replace the activation function with the approximate polynomials, the activation node $a_{i,j}$ is changed into $a_{i,j} + \Delta a_{i,j}$, where $\Delta a_{i,j}$ is a random variable representing an approximation error. These approximation errors $\{\Delta a_{i,j}\}$ introduce noise to $\mathcal{L}$, which we denote as $\Delta \mathcal{L} := \mathcal{L}(\{a_{i,j} + \Delta a_{i,j}\}) - \mathcal{L}(\{a_{i,j}\})$. By the first-order Taylor approximation, the noise in the loss function is given by $\Delta \mathcal{L} = \sum_{i,j} \frac{\partial \mathcal{L}}{\partial a_{i,j}} \Delta a_{i,j}$.

Suppose that we employ an identical approximate polynomial for each layer. For a given layer $i$, each individual approximation error $\Delta a_{i,j}$ is identically distributed since the nodes $\{a_{i,j} + \Delta a_{i,j}\}$ represent the output of the identical $i$th approximate polynomial. The variance of $\Delta a_{i,j}$ precisely aligns with $E_{\mu_i, \sigma_i^2}[d_i; f]$ as defined in Section 2. Here, $d_i$ denotes the degree of the $i$th approximate polynomial, while $\mu_i$ and $\sigma_i^2$ denote the mean and variance of the input data distribution, respectively. Furthermore, we assume that each random variable $\Delta a_{i,j}$ is independent. This assumption is reasonable given that each $\Delta a_{i,j}$ originates from independent polynomial approximations for each $i$th activation function, and each $a_{i,j}$ is computed independently using kernels for each $j$th activation node. Then, the variance of the loss function can be represented by

$$\text{Var}[\Delta \mathcal{L}] = \text{Var}\left[\sum_{i,j} \frac{\partial \mathcal{L}}{\partial a_{i,j}} \Delta a_{i,j}\right] = \sum_{i,j} \left(\frac{\partial \mathcal{L}}{\partial a_{i,j}}\right)^2 \text{Var}[\Delta a_{i,j}] = \sum_i \alpha_i E_{\mu_i, \sigma_i^2}[d_i; f], \quad (2)$$

where $\alpha_i = \sum_j \left(\frac{\partial \mathcal{L}}{\partial a_{i,j}}\right)^2$ quantifies the impact of the $i$th layer's approximation error on classification accuracy. Note that the values of $\alpha_i$ are different for each image. If we denote $\alpha_{i,k}$ as the value for the $k$th image in training set with $N_T$ images, then we can calculate the average of $\alpha_{i,k}$'s for all images in the training set, denoted as $A_i = \frac{1}{N_T} \sum_k \alpha_{i,k}$. Our objective is to minimize $\text{Var}[\Delta \mathcal{L}] = \sum_i A_i E_{\mu_i, \sigma_i^2}[d_i; f]$ by optimizing the set of degrees $\{d_1, \cdots, d_{N_L}\}$, which will be elaborated in the following subsection.

## 3.2 Optimization problem and dynamic programming approach

We formulate a layerwise degree optimization problem to minimize the variance of loss function $\text{Var}[\Delta \mathcal{L}]$ for a given constraint on inference time as follows:

$$\min_{d_1, \cdots, d_{N_L} \in \mathcal{S}} \sum_{i=1}^{N_L} A_i E_i[d_i; f] \qquad \text{subject to} \quad \sum_{i=1}^{N_L} T_i(d_i) \leq K, \quad (3)$$

where $K$ is a constraint on inference time, $N_L$ is the number of layers, and $\mathcal{S}$ is the degree search space. For simplicity, we represent $E_{\mu_i, \sigma_i}[\cdot; f]$ in Equation 2 as $E_i(\cdot)$. Then, we need to define the time required to perform the $i$th layer's bootstrapping and evaluate the approximate polynomial, denoted as $T_i(d_i)$. Note that $T_i(d_i)$ is an increasing function of $d_i$, which will be characterized in Section 4. Consequently, the total inference time becomes $C + \sum_{i=1}^{N_L} T_i(d_i)$, where $C$ denotes the runtime for other computations such as convolution operations. Since the choice of the approximate polynomial degrees $d_i$ does not affect other operations in deep neural networks, we do not need to consider the constant $C$ in the constraint on inference time.

Obtaining the optimal solution for this optimization problem is challenging due to the lack of a closed-form expression for $T_i(d_i)$. Hence, we introduce a discrete relaxation on $T_i(d_i)$ and convert $T_i(d_i)$ as $T_i^{\text{Rel}, \nu}(d_i) := \frac{1}{\nu} \lfloor T_i(d_i) \cdot \nu \rfloor$, where $\nu$ is an integer parameter, and $\lfloor \cdot \rceil$ is the rounding

function. With this relaxation, we tackle the optimization problem in Equation 3 via a dynamic programming approach.

In this approach, we decompose the optimization problem in Equation 3 into a set of simpler sub-problems $\{\mathcal{P}(l, k)\}$ as follows:

$$\min_{d_1, \cdots, d_l \in \mathcal{S}} \sum_{i=1}^{l} A_i E_i(d_i) \qquad \text{subject to} \qquad \sum_{i=1}^{l} T_i^{\text{Rel},\nu}(d_i) \leq k, \qquad (4)$$

where $l = 1, 2, \cdots, N_L$ and $k = \frac{1}{\nu}, \frac{2}{\nu}, \cdots, \frac{K \cdot \nu}{\nu}$. Let $\mathcal{D}(l, k) = (\mathcal{D}_1(l, k), \cdots, \mathcal{D}_l(l, k))$ represent the optimal solution for $\mathcal{P}(l, k)$, where $\mathcal{D}_i(l, k)$ denotes the degree of the $i$th polynomial for $i = 1, 2, \cdots, l$. Then, our main objective is to determine the optimal $\mathcal{D}(N_L, K)$, i.e., the solution of Equation 3. In some cases, there may be no degree pairs $d_1, \cdots, d_l$ that satisfy $\sum_{i=1}^{l} T_i^{\text{Rel},\nu}(d_i) \leq k$. To account for this situation, we set $T_i^{\text{Rel},\nu}(-1) := 0$, $E_i(-1) := \infty$ and include $-1$ in $\mathcal{S}$.

The main idea of the dynamic programming algorithm lies in the relation between $\mathcal{D}(l, \cdot)$ and $\mathcal{D}(l + 1, \cdot)$. The following theorem explicitly describes this relation.

**Theorem 3.1.** *For given $l = 1, 2, \cdots, N_L - 1$ and $k = \frac{1}{\nu}, \frac{2}{\nu}, \cdots, \frac{K \cdot \nu}{\nu}$, suppose that*

$$d' = \underset{d \in \mathcal{S}, \ T_{l+1}^{Rel,\nu}(d) \leq k}{\arg\min} \left[ \sum_{i=1}^{l} A_i E_i(\mathcal{D}_i(l, k - T_{l+1}^{Rel,\nu}(d))) + A_{l+1} E_{l+1}(d) \right].$$

*Then, $\mathcal{D}_i(l + 1, k) = \mathcal{D}_i(l, k - T_i^{Rel,\nu}(d'))$ for each $1 \leq i \leq l$, and $\mathcal{D}_{l+1}(l + 1, k) = d'$.*

Theorem 3.1 states that the optimal solution $\mathcal{D}(l + 1, k)$ of the problem $\mathcal{P}(l + 1, k)$ can be efficiently constructed if $\mathcal{D}(l, k')$ has been determined for all $k' = \frac{1}{\nu}, \frac{2}{\nu}, \cdots, \frac{K \cdot \nu}{\nu}$. We provide the proof of the theorem in Appendix C.

Based on this theorem, we construct a recursive algorithm to obtain $\mathcal{D}(N_L, K)$ as outlined below:

1. First, obtain $\mathcal{D}(l, k)$ for $l = 1$ and $k = \frac{1}{\nu}, \frac{2}{\nu}, \cdots, \frac{K \cdot \nu}{\nu}$. This can be easily achieved with $\mathcal{D}_1(1, k) = \max\{d \in \mathcal{S} | T_1^{\text{Rel},\nu}(d) \leq k\}$ as $E_1(\cdot)$ is minimized when $d_1$ is maximized.

2. For fixed $1 \leq l < N_L$, assume that $\mathcal{D}(l, k')$'s are already known for all $k' = \frac{1}{\nu}, \frac{2}{\nu}, \cdots, \frac{K \cdot \nu}{\nu}$. Then, for given $k$, find $d = d(k)$ which minimizes $\sum_{i=1}^{l} A_i E_i(\mathcal{D}_i(l, k - T_{l+1}^{\text{Rel},\nu}(d))) + A_{l+1} E_{l+1}(d)$ by brute-force searching through $d \in \mathcal{S}$. Here, the values of $E_i(\cdot)$ have to be pre-computed.

3. After finding $d(k)$ in the previous step for every $k = \frac{1}{\nu}, \frac{2}{\nu}, \cdots, \frac{K \cdot \nu}{\nu}$, we set $\mathcal{D}(l + 1, k)$ as $\mathcal{D}_i(l + 1, k) = \mathcal{D}_i(l, k - T_{l+1}^{\text{Rel},\nu}(d(k)))$ for $1 \leq i \leq l$, and $\mathcal{D}_{l+1}(l + 1, k) = d(k)$. This provides the optimal solution $\mathcal{D}(l + 1, k)$ for all $k$.

4. Continue to repeat steps 2 and 3 for $l = 1, 2, \cdots, N_L - 1$ until we obtain the optimal solution $\mathcal{D}(N_L, K)$.

---

**Algorithm 1** Dynamic programming algorithm for $\mathcal{P}(L, K)$

**Require:** $N_L, K, \mathcal{S}, \nu, E_i(\cdot), T_i^{\text{Rel},\nu}(\cdot)$
**Ensure:** $\mathcal{D}(N_L, K)$, the solution of $\mathcal{P}(N_L, K)$
1: **for** $k = \frac{1}{\nu}, \frac{2}{\nu}, \cdots, \frac{K \cdot \nu}{\nu}$ **do**
2: $\quad \mathcal{D}_1(1, k) \leftarrow \max\{d \in \mathcal{S} | T_1^{\text{Rel},\nu}(d) \leq k\}$
3: **end for**
4: **for** $l = 1, 2, \cdots, N_L - 1$ **do**
5: $\quad$ **for** $k = \frac{1}{\nu}, \frac{2}{\nu}, \cdots, \frac{K \cdot \nu}{\nu}$ **do**
6: $\quad\quad d' \leftarrow \arg\min_{d \in \mathcal{S}} \sum_{i=1}^{l} A_i E_i(\mathcal{D}_i(l, k - T_{l+1}^{\text{Rel},\nu}(d))) + A_{l+1} E_{l+1}(d)$
7: $\quad\quad$ **for** $i = 1, \cdots, l$ **do**
8: $\quad\quad\quad \mathcal{D}_i(l + 1, k) = \mathcal{D}_i(l, k - T_{l+1}^{\text{Rel},\nu}(d'))$
9: $\quad\quad$ **end for**
10: $\quad\quad \mathcal{D}_{l+1}(l + 1, k) = d'$
11: $\quad$ **end for**
12: **end for**

---

Note that the proposed algorithm can be executed with a time complexity $O(N_L K \nu |\mathcal{S}|)$, which is significantly more efficient than the brute-force searching through all degree search space, which requires a time complexity of $O(|\mathcal{S}|^{N_L})$. The procedure of this algorithm is summarized in Algorithm 1.

## 4 MODULI-CHAIN MANAGING

When approximating non-arithmetic operations with polynomials for private inference in deep neural networks on the RNS-CKKS scheme, previous works usually used the same polynomials for each layer. Thus, it is sufficient to use a single moduli-chain for implementing the deep neural network on FHE. However, the proposed polynomial approximation approach involves applying different polynomials with different degrees for each non-arithmetic operation by considering the characteristics of each layer. The single moduli-chain in the state-of-the-art scheme is designed to accommodate the maximum depth of the approximate polynomials. However, there is a problem of unused levels occurring in the single moduli-chain, when it is applied to the proposed approach. This makes the input data size to bootstrapping larger than optimal and thus noticeably increases the runtime. Therefore, we propose a method to optimize the bootstrapping runtime by the different moduli-chain for each layer as the different depth is used for each activation function. In this section, we propose a method to reduce the runtime of the bootstrapping by removing unused moduli based on the specific requirements of each layer, thus optimizing the situation for each layer accordingly. [2]

Previous works designed the moduli-chain for private inference on deep neural networks as follows. The modulus of the ciphertext is denoted as $Q_L = \prod_{i=0}^{L} q_i$. Here, $q_0$ represents the base modulus, and $q_1, \cdots, q_\delta$ correspond to the moduli required for deep neural network operations that consume depth, such as convolutional layers or approximate activation polynomials. The remaining $q_{\delta+1}, \cdots, q_L$ are moduli for bootstrapping, each of which is far larger than $q_1, \cdots, q_\delta$. When performing private inference, we use moduli $q_1, \cdots, q_\delta$ to evaluate homomorphic operations. Then, when the level of the ciphertext becomes 0, we raise the level of the ciphertext to $l$, and perform bootstrapping by consuming $l - \delta$ levels using the bootstrapping moduli $q_{\delta+1}, \cdots, q_L$. Let $l_{\max}$ be the maximum depth of the approximate polynomial for the activation function among all layers in a ResNet model, and let $l_{\text{conv}}$ be the depth of the convolution operation. Then we have $l_{\text{conv}} + l_{\max} = \delta$. If we want to evaluate a layer with approximate polynomial with the depth $l < l_{\max}$ with this moduli-chain, then the bootstrapping operation is performed in the level using the bootstrapping moduli $q_{\delta+1}, \cdots, q_L$, and we drop the $l_{\max} - l$ levels before evaluating activation function and the convolution. Then, the runtime of the bootstrapping is the same regardless of the depth of the activation function.

Instead of using a single moduli-chain, we propose multiple moduli-chains for each depth of the activation function. If a layer has an approximate polynomial for activation function with the depth $l < l_{\max}$, the moduli $q_0, q_1, \cdots, q_{l_{\text{conv}}+l}, q_{\delta+1}, \cdots, q_L$ are enough for evaluation moduli in the moduli chain for the layer. Then, the bootstrapping operation of this case does not have to operate for moduli $q_{l_{\text{conv}}+l}, \cdots, q_\delta$. It reduces the runtime of the bootstrapping. We can choose which moduli-chain is used for the next layer when the ciphertext level is zero and should be raised with the modulus for the next bootstrapping. Since the coefficients of the secret key are from only $\{-1, 0, 1\}$, the secret key is independent of the moduli, and we can use the several moduli-chains sequentially for one ciphertext encrypted with one secret key.

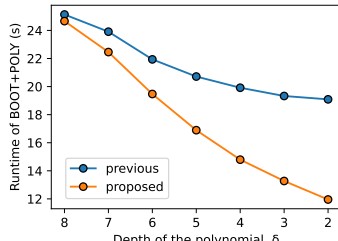

Figure 1: Comparison of runtime between maintaining the modulus and moduli-chain managing methods, the increasing function of the depth consumption $\delta$.

**Simulation results.** We validate the amount of time saved through simulation when removing unused moduli during the bootstrapping process in the proposed operations. Figure 1 shows the difference of runtime for single bootstrapping and the proposed approximate polynomial evaluation, $T_i(d_i)$ in Equation 3. Considering the largest degree that consumes the level $\delta$ is $2^\delta - 1$, we perform the approximate polynomials for the degree $d_i = 2^\delta - 1$ with $\delta = 2, 3, \cdots, 8$. Also we set $l_{\text{conv}} = 2$, $l_{\max} = 8$, and $l_{\text{Boot}} = 14$. For each $\delta$, we compare the runtime for bootstrapping and the following polynomial evaluation consuming depth of $\delta$. The result of our simulation shows that removing unused moduli boosts the time of bootstrapping and evaluating approximate polynomials significantly.

---

[2]Preliminaries about the RNS-CKKS scheme and the bootstrapping are described in Appendix A.

## 5 EXPERIMENTAL RESULTS

In this section, we provide the experimental results, which show the performance of the layerwise optimized approximate polynomials by the proposed algorithm. We simulated the classification task of the CIFAR-10 dataset (Krizhevsky et al., 2009) with the pre-trained backbone network ResNet (He et al., 2016) and compared the proposed method with the previous approximation method. Also, we performed the encryption and homomorphic operations of the RNS-CKKS by the `Lattigo` library. The `Lattigo` library is faster than the `SEAL` library with bootstrapping used in Lee et al. (2022a) due to the optimized bootstrapping. Therefore, we repeat the previous works in the `Lattigo` library for a fair comparison rather than simply referring to the reported runtime in the previous works. We simulate every experiment on AMD Ryzen Threadripper PRO 3995WX at 2.096 GHz (64 cores) with 512 GB RAM, running the Ubuntu 20.04 operating system.

Before providing results, we discuss the role of the coefficient $\lambda$ in Section 2.3 on the practical usage. For the ResNet-32 backbone network used in CIFAR-10 classification, we approximate the ReLU function of the first layer as the polynomial degree $d = 255$, with $\mu = 0.3165$, $\sigma = 0.8785$, $B_1 = -11.2692$, $B_2 = 12.3143$, and $r = 2.7$, which are defined in Section 2. The maximum absolute value of the coefficients in the decomposed polynomials is $6.59 \times 10^9$, which is not proper for performing stable polynomial evaluation. By adopting a proper $\lambda$, we can suppress the maximum coefficient value effectively, as shown in Figure 2. When the absolute value of a coefficient of decomposed polynomials exceeds $10^6$, we apply $\lambda = 2$ to that layer.

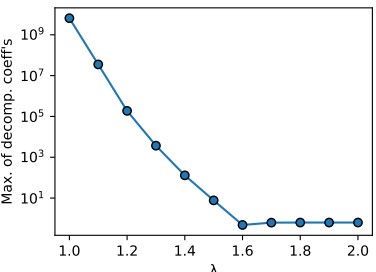

Figure 2: The maximum of absolute values of coefficients of decomposed polynomials with respect to $\lambda$.

### 5.1 RESULTS OF DYNAMIC PROGRAMMING AND COMPARISON OF PLAINTEXT RESULTS

Table 1: The result of dynamic programming in Algorithm 1 solving the optimization problem and its classification accuracy for the plaintext CIFAR-10 dataset. The asterisks(*) means the polynomial in which $\lambda = 2$ is applied.

| Backbone | Degrees for each layer | Accuracy |
|---|---|---|
| ResNet-20 (91.48%) | 63,31,31*,31,31,31,31*,31,31,15,31,15,15,31,31,31,31,15,15 | 90.14% |
| ResNet-32 (92.49%) | 255*,63,63,63,63,31,63,31,31,31,31,63,15,63,15,63,15,63,31,63,31,63,31,63,31,31,31,31,15,7 | 91.78% |

We use the pre-trained ResNet-20 and ResNet-32 for the CIFAR-10 dataset. In ResNet models, all non-arithmetic functions are ReLU. We scale $\sigma$ with $r = 2.2, 2.7$ for ResNet-20, 32, respectively. We use the degree search space $\mathcal{S} = \{-1, 7, 15, 31, 63, 127, 255\}$ and the discrete relaxation parameter as $\nu = 4$ for our optimization. Each element of the degree search space $\mathcal{S}$ corresponds to the polynomial degree that minimizes the MSE for each depth, specifically defined as $2^\delta - 1$ for each depth $\delta$. For finer control over the polynomial degrees, the search space can include more polynomial degrees. The optimized degrees are presented in Table 1. The proposed method can construct accurate models with reduced polynomial degrees while incurring marginal reduction in classification accuracy.

Table 2: The accuracy and degree comparisons for performing plaintext inference for the CIFAR-10 on ResNet models between the previous method and the proposed optimization method.

| Backbone | Method | Degrees | Max. depth. | Original | Approx. | Diff. (%) |
|---|---|---|---|---|---|---|
| ResNet-20 | Minimax | All degrees are 6,705 | 14 | 88.36% | 88.01% | -0.35 |
| | | All degrees are 2,835 | 13 | 88.36% | 86.44% | -1.92 |
| | Proposed | {63,31,31,...} | 6 | 91.48% | 90.71% | -0.77 |
| ResNet-32 | Minimax | All degrees are 6,705 | 14 | 89.38% | 88.99% | -0.39 |
| | | All degrees are 2,835 | 13 | 89.38% | 88.08% | -1.30 |
| | Proposed | {255,63,63,...} | 8 | 92.49% | 91.65% | -0.84 |

We compare the proposed optimization method with the previous minimax approximation in Table 2. Lee et al. (2023a) proposed the method approximating the ReLU by minimax polynomial

approximation with various levels of precision and reported its performance in plaintext inference. For the ResNet-20, if we reduce the depth only by one using the minimax approximation, the classification accuracy decreases by 1.92%. However, the proposed method can reduce the depth by 8 with only 0.77% loss in the accuracy, which highlights the effectiveness of the promoted method.

## 5.2 RESULTS OF CIPHERTEXT RESULTS

We use the implementation in Lee et al. (2022a) as our baseline to validate the proposed method empirically. Our approximation techniques for activation functions are independent of the convolution operation for FHE and can be applied to alternative implementations (e.g., Kim & Guyot (2023)) for similar benefits.

We set the degree of polynomial for ciphertexts $N = 16$. As the previous work used the total modulus bit length $\log(PQ) = 1553$ with the secret key of Hamming weight $h = 192$ for satisfying the 128-bit security, we also optimized the modulus bit length of the proposed moduli-chain. In the same way as the previous work, we set base modulus, bootstrapping moduli, and special moduli as 51-bit primes and other evaluation moduli as 46-bit primes. For each implementation, the number of special moduli is determined to satisfy 128-bit security while achieving the fastest key switching time. For other parameters, such as the Hamming weight of the secret key and bootstrapping parameters, we follow the settings used in Lee et al. (2022a).

Table 3: The runtime comparison for performing private inference of single image on the ResNet-20 between the previous method and the proposed method.

| Backbone | Method | Max. deg. | Max. depth. | Boot.(s) | Poly.(s) | Total(s) |
|---|---|---|---|---|---|---|
| ResNet-20 | (Lee et al., 2022a) | 6,075 | 14 | 936 | 164 | 1,169 |
| | Proposed | 63 | 6 | 240 | 33 | **339** |
| ResNet-32 | (Lee et al., 2022a) | 6,075 | 14 | 1,552 | 268 | 1,922 |
| | Proposed | 255 | 8 | 430 | 71 | **607** |

Under the same condition of 128-bit security, we compare the previous state-of-the-art approximation method and the proposed method to show how much inference runtime is saved by the proposed method. Table 3 shows the results. By optimizing the maximum degree of the approximate polynomials with the weighted least squares method, we can reduce the runtime of each key-switching operation by increasing the number of the special moduli, which leads to reduce the number of operations in each bootstrapping and each polynomial. Additionally, optimizing each layer's degree via the dynamic programming further reduces the inference runtime. The proposed optimization leads to a substantial reduction in total inference time by 3.44 times for the ResNet-20 model and 3.16 times for the ResNet-32 model compared to using the minimax approximation. Instead of using the presented time records in Lee et al. (2022a), we simulated the previous work again in the same environment for a fair comparison. Thus, the baseline in Table 3 is also faster than the previous results.

Finally, we classify all of the encrypted test dataset of CIFAR-10 with the proposed approximate polynomials on the ResNet-20 and 32 models, and Table 4 shows the result. We confirm that the proposed method can construct accurate models with reduced polynomial degrees while incurring marginal reduction in classification accuracy for ciphertext also.

Table 4: The classification accuracy of ResNet models on the homomorphically encrypted CIFAR-10 by the Lattigo library.

| Backbone | Original | Plaintext | Ciphertext |
|---|---|---|---|
| ResNet-20 | 91.48% | 90.14% | 90.62% |
| ResNet-32 | 92.49% | 91.78% | 90.85% |

## 6 CONCLUSION

We proposed the weighted least squares method and the dynamic programming algorithm for layerwise optimization of approximate polynomial degrees, along with a moduli-chain management technique to reduce bootstrapping time. Our methods achieved substantial improvements in inference time for the ResNet-20 model and the ResNet-32 model, achieving speedups of 3.44 times and 3.16 times, respectively, compared to the existing state-of-the-art method.

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

## A  PRELIMINARIES: RNS-CKKS SCHEME AND BOOTSTRAPPING

The FHE scheme is an encryption scheme that supports operations on encrypted data, and the residue number system variant of Cheon-Kim-Kim-Song (RNS-CKKS) scheme (Cheon et al., 2018) is one of the prominent FHE schemes that support operations for real (or complex) number data. The RNS-CKKS scheme encrypts real (or complex) number data stored in the form of a 1-dimensional vector into the form $(b, a) \in \mathcal{R}_{Q_l}^2$, where $Q_l := \prod_{i=0}^{l} q_i$ is the product of prime numbers $q_i$, and $\mathcal{R}_{Q_l} := \mathbb{Z}_{Q_l}[X]/\langle X^N + 1 \rangle$. The primes $q_i$ are used to represent the coefficients of elements in $\mathcal{R}_{Q_l}^2$ as a residual number system to accelerate operations among ciphertexts. When a ciphertext is in $\mathcal{R}_{Q_l}^2$, its level is referred to as $l$ ($l \geq 0$), where the ciphertext level refers to computational resources that define the possible number of homomorphic multiplications.

There are five primary homomorphic operations in the RNS-CKKS scheme: addition, multiplication with plaintext, multiplication with ciphertext, rotation, and conjugation. Among these, the homomorphic multiplication with a plaintext or a ciphertext decreases the ciphertext level by one. When the ciphertext reaches level 0, it is no longer possible to perform homomorphic multiplication. Therefore, the server receiving the ciphertext is restricted in using homomorphic multiplication without decryption or the assistance of a third party. However, through bootstrapping, the level of the ciphertext can be increased, allowing further homomorphic multiplications to be performed.

When we evaluate the polynomial for ciphertexts, we perform a baby-step giant-step (BSGS) algorithm to evaluate the polynomial with optimal depth $\lceil \log_2(d + 1) \rceil$. (Bossuat et al., 2021) The BSGS algorithm decomposes the polynomial as a quotient polynomial and a remainder polynomial dividing the Chebyshev polynomials. We proactively store the coefficients of those decomposed polynomials in the tree structure. The BSGS algorithm evaluates Chebyshev polynomials from the input values as a baby-step, and combines them using the pre-stored coefficients as a giant-step. The one thing we have to consider for performing the BSGS algorithm is that the magnitudes of the coefficients the algorithm stores in the tree should not be excessively large.

Bootstrapping is the process of transforming a ciphertext $\mathsf{ct}$ at level 0, denoted as $\mathsf{ct} \in \mathcal{R}_{Q_0}^2$, into a ciphertext $\mathsf{ct}_{\text{Boot}}$ at level $l' > 0$, represented as $\mathsf{ct}_{\text{Boot}} \in \mathcal{R}_{Q_{l'}}^2$, while ensuring that the decryption results of $\mathsf{ct}$ and $\mathsf{ct}_{\text{Boot}}$ are the same with some approximation noise. To achieve this, we first raise the level of $\mathsf{ct}$ to $L_{\text{Boot}}$, irrespective of the decryption result. The resulting ciphertext $\mathsf{ct}'$ will have a decryption result in the form of $\Delta \cdot m + q_0 \cdot I$ for some integer coefficient polynomial $I$, where $m$ is the message polynomial, decryption of $\mathsf{ct}$. Subsequently, we perform modular reduction to eliminate $q_0 \cdot I$. During this process, the level of the ciphertext is reduced by $l_{\text{Boot}}$, resulting in a ciphertext with a level of $l' = L_{\text{Boot}} - l_{\text{Boot}}$. As a result, the decryption result of the final ciphertext closely matches the decryption result of $\mathsf{ct}$ while having a level of $l'$. To perform multiple operations among ciphertexts using FHE, it is essential to carefully design the entire operation process, taking into consideration factors such as bootstrapping and the consumption of levels. According to the results of deep neural network operations in the previously conducted research using the RNS-CKKS scheme, bootstrapping consumes more than 70% of the overall time (Lee et al., 2022a). Therefore, while it is possible to use bootstrapping as much as needed to continue homomorphic multiplication, it comes at the cost of significant time consumption.

## B  PROOF OF THEOREMS IN SECTION 2

In this section, we provide the proof of the theorems in Section 2.

We recall the Hilbert space theory in mathematical analysis to approach the approximation by the weight least squares method.

**Lemma B.1** (Hilbert Projection Theorem). *Let $H$ be a Hilbert space, $\| \cdot \|_H$ be a norm defined on $H$, and $C$ be a nonempty closed convex set contained in $H$. Let $h \in H$ be a vector in the Hilbert space. Then, there exists a unique vector $c \in C$ such that*

$$\|h - c\|_H \leq \|h - c'\|_H$$

*for any $c' \in H$.*

This lemma asserts that the *closest* vector through $C$ from $h \in H$ always exists and is unique. Now, we consider a function space $L(\mathbb{R}, \phi) = \{f : \mathbb{R} \to \mathbb{R} | \int_{\mathbb{R}} \phi(x) f(x)^2 dx < \infty\}$, where

the inner product $\langle \cdot, \cdot \rangle_\phi$ is defined as $\langle f, g \rangle_\phi := \int_\mathbb{R} \phi(x) f(x) g(x) dx$, and the norm as $\|f\|_\phi := (\int_\mathbb{R} \phi(x) f(x)^2 dx)^{\frac{1}{2}}$. Here, the function $\phi : \mathbb{R} \to \mathbb{R}^{\geq 0}$ is a weight function and always has a non-negative value. Then, the space $L(\mathbb{R}, \phi)$ is a Hilbert space. Also, note the set

$$C_d := \{p(x) : \text{ polynomial} \mid p(x) \in L(\mathbb{R}, \phi), \text{ the degree of } p(x) \text{ is not greater than } d\}$$

is closed set in $L(\mathbb{R}, \phi)$ for every integer $d \geq 0$. Therefore, Lemma B.1 can be applied as follows:

**Theorem B.1.** *Let $f \in L(\mathbb{R}, \phi)$. Then, for each non-negative integer $d$, there exists a polynomial $P[d](x)$ with degree not greater than $d$ which satisfies*

$$\int_\mathbb{R} \phi(x)(f(x) - P[d](x))^2 dx \leq \int_\mathbb{R} \phi(x)(f(x) - q(x))^2 dx$$

*for any polynomial $q(x)$ with degree not greater than $d$.*

Theorem B.1 is just a re-statement of Theorem 2.1. From now on, we define a polynomial $P[d; f](x)$ to be the closest vector from $f \in L(\mathbb{R}, \phi)$.

Now, we provide the proof of Theorem 2.2, which states the form of $P[d; f](x)$ when the weight function is a normal distribution, $\phi(x) = \frac{1}{\sqrt{2\pi\sigma^2}} e^{-\frac{(x-\mu)^2}{2\sigma^2}}$.

*Proof of Theorem 2.2.* We define a polynomial

$$h_n(x) := \frac{(-1)^n}{\sqrt{n!}} e^{x^2/2} \frac{d^n}{dx^n} e^{-x^2/2}, \tag{5}$$

a polynomial with degree $n$. The polynomial $h_n$ has the property

$$\int_\mathbb{R} \frac{1}{\sqrt{2\pi}} e^{-\frac{1}{2}z^2} h_m(z) h_n(z) dz = \delta_{mn},$$

where $\delta_{mn}$ is Kronecker delta function.

Note that any polynomial $p(x) \in C_d$, which degree is not greater than $d$, can be expressed as

$$p(x) = b_0 h_0 \left( \frac{x - \mu}{\sigma} \right) + b_1 h_1 \left( \frac{x - \mu}{\sigma} \right) + \cdots + b_d h_d \left( \frac{x - \mu}{\sigma} \right)$$

for $b_j \in \mathbb{R}$, $j = 0, 1, \cdots, d$. Then, the norm of the difference between the function $f$ and the polynomial $p$ is

$$\|f - p\|_\phi = \left( \int_\mathbb{R} \phi(x) \left( f(x) - \sum_j b_j h_j \left( \frac{x - \mu}{\sigma} \right) \right)^2 dx \right)^{\frac{1}{2}}.$$

If we find the real numbers $b_0, \cdots, b_d$ which minimizes $\|f - p\|_\phi$, then we can conclude that the polynomial $b_0 h_0 \left( \frac{x-\mu}{\sigma} \right) + b_1 h_1 \left( \frac{x-\mu}{\sigma} \right) + \cdots + b_d h_d \left( \frac{x-\mu}{\sigma} \right)$ is the same as $P[d; f](x)$. Some algebra

shows that

$$
\begin{aligned}
\frac{\partial}{\partial b_n}\|f-p\|_\phi^2 &= \frac{\partial}{\partial b_n}\int_{\mathbb{R}}\phi(x)\left(f(x)-\sum_j b_j h_j\left(\frac{x-\mu}{\sigma}\right)\right)^2 dx \\
&= \frac{\partial}{\partial b_n}\int_{\mathbb{R}}\frac{1}{\sqrt{2\pi}}e^{-\frac{1}{2}z^2}\left(f(\mu+\sigma z)-\sum_j b_j h_j(z)\right)^2 dz \\
&= \frac{\partial}{\partial b_n}\left[\int_{\mathbb{R}}\frac{1}{\sqrt{2\pi}}e^{-\frac{1}{2}z^2}f(\mu+\sigma z)^2 dz + \sum_{i,j}b_i b_j\int_{\mathbb{R}}\frac{1}{\sqrt{2\pi}}e^{-\frac{1}{2}z^2}h_i(z)h_j(z)\right. \\
&\qquad\qquad\left. -2\sum_j b_j\int_{\mathbb{R}}\frac{1}{\sqrt{2\pi}}e^{-\frac{1}{2}z^2}f(\mu+\sigma z)h_j(z)dz\right] \\
&= 0 + \frac{\partial}{\partial b_n}\left[\sum_j b_j^2 - 2\sum_j b_j\int_{\mathbb{R}}\frac{1}{\sqrt{2\pi}}e^{-\frac{1}{2}z^2}f(\mu+\sigma z)h_j(z)dz\right] \\
&= 2b_n - 2\int_{\mathbb{R}}\frac{1}{\sqrt{2\pi}}e^{-\frac{1}{2}z^2}f(\mu+\sigma z)h_j(z)dz.
\end{aligned}
$$

Therefore, $\frac{\partial}{\partial b_n}\|f-p\|_\phi^2 = 0$ leads to $b_n = \int_{\mathbb{R}}\frac{1}{\sqrt{2\pi}}e^{-\frac{1}{2}z^2}f(\mu+\sigma z)h_j(z)dz$ for all $n = 0,\cdots,d$, and this implies

$$
P[d;f](x) = \sum_j h_j\left(\frac{x-\mu}{\sigma}\right)\int_{\mathbb{R}}\frac{1}{\sqrt{2\pi}}e^{-\frac{1}{2}z^2}f(\mu+\sigma z)h_j(z)dz,
$$

which is equivalent to

$$
P[d;f](x) = \sum_j h_j\left(\frac{x-\mu}{\sigma}\right)\int_{\mathbb{R}}\phi(t)f(t)h_j\left(\frac{t-\mu}{\sigma}\right)dt.
$$

Now, we induce the MSE from the $P[d;f]$, the approximation of $f$. The MSE refers to $\|f - P[d;f]\|_\phi^2$, the square of the norm of the difference between $f$ and the polynomial $P[d;f]$.

Let $\eta_j(t) := h_j\left(\frac{t-\mu}{\sigma}\right)$. Then, $\langle\eta_i,\eta_j\rangle_\phi = \delta_{ij}$, and $\int_{\mathbb{R}}\phi(t)f(t)h_j\left(\frac{t-\mu}{\sigma}\right)dt$ can be expressed as $\langle f,\eta_j\rangle_\phi$. Note that

$$
\langle f, P[d;f]\rangle_\phi = \left\langle f, \sum_j \eta_j\langle f,\eta_j\rangle_\phi\right\rangle_\phi = \sum_j\langle f,\eta_j\rangle_\phi^2,
$$

$$
\begin{aligned}
\langle P[d;f], P[d;f]\rangle_\phi &= \left\langle \sum_j \eta_j\langle f,\eta_j\rangle_\phi, \sum_j \eta_j\langle f,\eta_j\rangle_\phi\right\rangle_\phi \\
&= \sum_{i,j}\langle f,\eta_i\rangle_\phi\langle f,\eta_j\rangle_\phi\langle\eta_i,\eta_j\rangle_\phi = \sum_j\langle f,\eta_j\rangle_\phi^2,
\end{aligned}
$$

which imply

$$
\begin{aligned}
\|f - P[d;f]\|_\phi^2 &= \langle f - P[d;f], f - P[d;f]\rangle_\phi \\
&= \langle f,f\rangle_\phi - 2\langle f, P[d;f]\rangle_\phi + \langle P[d;f], P[d;f]\rangle_\phi \\
&= \langle f,f\rangle_\phi - \sum_j\langle f,\eta_j\rangle_\phi^2.
\end{aligned}
$$

Therefore, we have

$$
\begin{aligned}
E_{\mu,\sigma^2}[d;f] &= \int_{\mathbb{R}}\phi(x)(f(x)-P[d;f](x))^2 dx \\
&= \int_{\mathbb{R}}\phi(x)f(x)^2 dx - \sum_{j=0}^d\left[\int_{\mathbb{R}}\phi(x)f(x)h_j\left(\frac{x-\mu}{\sigma}\right)dx\right]^2,
\end{aligned}
$$

and this proves Theorem 2.2. $\qquad\square$

## C    PROOF OF THEOREM IN SECTION 3.2

In this section, we prove that the proposed algorithm gives degree set $\{d_1, \cdots, d_{N_L}\}$, which solves the optimization problem. We recall the notations that we defined in Section 3.2. Let $\mathcal{P}(l, k)$ be the optimization problem in Equation 4,

$$\min_{d_1, \cdots, d_l} \sum_{i=1}^{l} A_i E_i(d_i) \qquad \text{subject to} \qquad \sum_{i=1}^{l} T_i^{\text{Rel},\nu}(d_i) \leq k,$$

where $l = 1, 2, \cdots, N_L$ and $k = \frac{1}{\nu}, \frac{2}{\nu}, \cdots, \frac{K \cdot \nu}{\nu}$. Recall that the MSE function $E_i(d)$ is a decreasing function with respect to $d$, and the runtime function $T_i^{\text{Rel},\nu}(d)$ is discrete, a multiple of $\frac{1}{\nu}$.

Let $\mathcal{D}(l, k)$ be the optimal solution of $\mathcal{P}(l, k)$, and $\mathcal{D}_i(l, k)$ be the degree of the $i$th polynomial in the optimal solution $\mathcal{D}(l, k)$. Now we prove the relation between $\mathcal{D}(l, \cdot)$ and $\mathcal{D}(l + 1, \cdot)$, which we state in Theorem 3.1.

*Proof.* To show $\mathcal{D}_i(l+1, k) = \mathcal{D}_i(l, k - T_{l+1}^{\text{Rel},\nu}(d'))$ for $i = 1, \cdots, l$ and $\mathcal{D}_{l+1}(l+1, k) = d'$ where

$$d' = \arg\min_{d \in \mathcal{S}, \, T_{l+1}^{\text{Rel},\nu}(d) \leq k} \left[ \sum_{i=1}^{l} A_i E_i(\mathcal{D}_i(l, k - T_{l+1}^{\text{Rel},N}(d))) + A_{l+1} E_{l+1}(d) \right],$$

we have to check that

$$\sum_{i=1}^{l} A_i E_i(\mathcal{D}_i(l, k - T_{l+1}^{\text{Rel},\nu}(d'))) + A_{l+1} E_{l+1}(d') \leq \sum_{i=1}^{l+1} A_i E_i(d_i) \qquad (6)$$

for any $d_i \in \mathcal{S}$ which satisfies $\sum_{i=1}^{l+1} T_i^{\text{Rel},\nu}(d_i) \leq k$. Since $\sum_{i=1}^{l} T_i^{\text{Rel},\nu}(d_i) \leq k - T_{l+1}^{\text{Rel},\nu}(d_{l+1})$,

$$\sum_{i=1}^{l} A_i E_i(\mathcal{D}_i(l, k - T_{l+1}^{\text{Rel},\nu}(d_{l+1}))) \leq \sum_{i=1}^{l} A_i E_i(d_i) \qquad (7)$$

holds for any $d_i \in \mathcal{S}$, from the optimality of $\mathcal{D}(l, k - T_{l+1}^{\text{Rel},\nu}(d_{l+1}))$. Also, from the definition of $d'$,

$$\begin{aligned} &\sum_{i=1}^{l} A_i E_i(\mathcal{D}_i(l, k - T_{l+1}^{\text{Rel},\nu}(d'))) + A_{l+1} E_{l+1}(d') \\ &\leq \sum_{i=1}^{l} A_i E_i(\mathcal{D}_i(l, k - T_{l+1}^{\text{Rel},\nu}(d_{l+1}))) + A_{l+1} E_{l+1}(d_{l+1}) \end{aligned} \qquad (8)$$

for any $d_{l+1} \in \mathcal{S}$. Combining Equation 7 and Equation 8 leads to Equation 6, and this proves the theorem. $\qquad\square$

## D    ADJUSTMENT OF THE COEFFICIENTS FOR APPROXIMATE POLYNOMIALS

In this section, we propose the method of adjustment of the approximate polynomial for stable polynomial evaluation in the ciphertext domain.

Unlike in the plaintext domain, it is essential that input values fall within the interval $[-1, 1]$ to enable stable operations in the ciphertext domain. To direct input values to fall within the range of $[-1, 1]$, we define the *evaluation region* for each activation function. Suppose that all input values of the $i$th activation function $f(x)$ are fall in the region $[B_{1,i}, B_{2,i}]$ for the $i$th activation function $f(x)$. Then, instead of approximating $f(x)$, we approximate the function $\tilde{f}(z) = f(\frac{B_{2,i} - B_{1,i}}{2} z + \frac{B_{2,i} + B_{1,i}}{2})$ to force the input values as $z \in [-1, 1]$, while the output values are the same. Shifting the region $[B_{1,i}, B_{2,i}]$ to $[-1, 1]$, the input distribution also changes from the normal distribution $\mathcal{N}(\mu_i, \sigma_i^2)$

to $\mathcal{N}(\tilde{\mu}_i, \tilde{\sigma}_i^2)$, where $\tilde{\mu}_i = (\mu_i - \frac{B_{2,i}+B_{1,i}}{2})/(\frac{B_{2,i}-B_{1,i}}{2})$, and $\tilde{\sigma}_i = \sigma_i/(\frac{B_{2,i}-B_{1,i}}{2})$. Then, one can use the approximate polynomial $P_{\tilde{\mu}_i,\tilde{\sigma}_i^2}[d_i; \tilde{f}](z)$ for $z \in [-1, 1]$, instead of $P_{\mu_i,\sigma_i^2}[d_i; f](x)$ for $x \in [B_{1,i}, B_{2,i}]$.

One of the main problems from this approximation is that the parameters for evaluation region $B_{1,i}$ and $B_{2,i}$ affect the multipliers. When we approximate the function on a wide evaluation region, the coefficients of the approximate polynomial grow rapidly when we set the length of the evaluation region wider and fix the center of the region.

To deal with this issue, we adjust the approximation to reduce the magnitudes of the multipliers while using the same evaluation region. More specifically, we modify the polynomial $P_{\tilde{\mu}_i,\tilde{\sigma}_i^2}[d_i; \tilde{f}](z)$ as $P_{\tilde{\mu}_i,(\lambda_i\tilde{\sigma}_i)^2}[d_i; \tilde{f}](z)$ for some constant $\lambda_i > 1$. Since the normal distribution $\mathcal{N}(\tilde{\mu}_i, (\lambda_i\tilde{\sigma}_i)^2)$ is also concentrated near $\tilde{\mu}_i$ like $\mathcal{N}(\tilde{\mu}_i, \tilde{\sigma}_i^2)$, We can expect that making slight changes to the variance of the input distribution will not significantly impact the inference. We show that the constant $\lambda_i$ may reduce the magnitude of the high-degree coefficient of the polynomial $P_{\tilde{\mu}_i,(\lambda_i\tilde{\sigma}_i)^2}[d_i; \tilde{f}](z)$ rapidly, under the condition of the function $f(x)$ which can be bounded by some linear function. Commonly used activation functions are mostly bounded by linear functions, so this assumption can be applied in general.

We summarized the above discussion in the theorem below.

**Theorem D.1.** Let $\sum_{l=0}^{d} a_l z^l$ be the approximate polynomial $P_{\tilde{\mu},\tilde{\sigma}^2}[d; \tilde{f}](z)$, where $\tilde{f}(z) = f(\frac{B_2-B_1}{2}z + \frac{B_2+B_1}{2})$. If $\mu$ and $\sigma^2$ are given, and $\frac{B_1+B_2}{2}$ is fixed, then there exists a constant $D_l$ independent of the choice of $B_1$ and $B_2$, such that $a_l = D_l(B_2 - B_1)^l$.

**Theorem D.2.** Let $\sum_{l=0}^{d} a_l z^l$ be the approximate polynomial $P_{\tilde{\mu},(\lambda\tilde{\sigma})^2}[d; \tilde{f}](z)$, where $\tilde{f}(z) = f(\frac{B_2-B_1}{2}z + \frac{B_2+B_1}{2})$. If the function $f(x)$ can be bounded by some linear function (i.e., there exists a constant $\beta$ and $\gamma$ such that $|f(x)| < |\beta x + \gamma|$ for every $x$), then for any $l \geq 3$, $|a_l| \leq C_l/\lambda^{l-2}$ for fixed constant $C_l$.

We provide the proof of the above theorems in this section. Before starting the proof, we propose two lemmas.

**Lemma D.1.** Let $\tilde{f}(z) = f(\frac{B_2-B_1}{2}z + \frac{B_2+B_1}{2})$. Then, for $\tilde{\mu} = (\mu - \frac{B_2+B_1}{2})/(\frac{B_2-B_1}{2})$ and $\tilde{\sigma} = \sigma/(\frac{B_2-B_1}{2})$,

$$P_{\tilde{\mu},\tilde{\sigma}^2}[d; \tilde{f}](z) = P_{\mu,\sigma^2}[d; f]\left(\frac{B_2 - B_1}{2}z + \frac{B_2 + B_1}{2}\right).$$

*Proof.* By definition, the polynomial $q(z) := P_{\tilde{\mu},\tilde{\sigma}^2}[d; \tilde{f}](z)$ is the unique polynomial (through the polynomials with degree less than $d$) which minimizes

$$\|\tilde{f} - q\|_{\tilde{\phi}} = \int_{\mathbb{R}} \tilde{\phi}(z)(\tilde{f}(z) - q(z))^2 dz,$$

where $\tilde{\phi}(x) = \frac{1}{\sqrt{2\pi\tilde{\sigma}^2}}e^{-\frac{(x-\tilde{\mu})^2}{2\tilde{\sigma}^2}}$. Substituting $x := \frac{B_2-B_1}{2}z + \frac{B_2+B_1}{2}$, $\tilde{\mu} := \frac{B_2-B_1}{2}\mu + \frac{B_2+B_1}{2}$,

$$\int_{\mathbb{R}} \tilde{\phi}(z)(\tilde{f}(z) - q(z))^2 dz = \int_{\mathbb{R}} \tilde{\phi}\left(\frac{x - \frac{B_2+B_1}{2}}{\frac{B_2-B_1}{2}}\right)\left[f(x) - q\left(\frac{x - \frac{B_2+B_1}{2}}{\frac{B_2-B_1}{2}}\right)\right]^2 \frac{2}{B_2 - B_1} dx$$

$$= \int_{\mathbb{R}} \phi(x)\left[f(x) - q\left(\frac{x - \frac{B_2+B_1}{2}}{\frac{B_2-B_1}{2}}\right)\right]^2 dx,$$

where $\phi(x) = \frac{1}{\sqrt{2\pi\sigma^2}}e^{-\frac{(x-\mu)^2}{2\sigma^2}}$. Therefore, the polynomial $q\left((x - \frac{B_2+B_1}{2})/(\frac{B_2-B_1}{2})\right)$ equals to $P_{\mu,\sigma^2}[d; f](x)$. Substituting again, $P_{\tilde{\mu},\tilde{\sigma}^2}[d; \tilde{f}](z) = q(z) = P_{\mu,\sigma^2}[d; f](\frac{B_2-B_1}{2}z + \frac{B_2+B_1}{2})$. $\square$

**Lemma D.2.** Let $\sum_{l=0}^{d} a_l z^l$ be a polynomial $P_{\tilde{\mu},\tilde{\sigma}^2}[d; \tilde{f}](z)$, where the notations $\tilde{\mu}$, $\tilde{\sigma}$, and $\tilde{f}$ follow Lemma D.1 for given paramters $B_1$ and $B_2$. Then,

$$a_l = \frac{1}{\sqrt{l!}}\sum_{j=l}^{d}\left[\binom{j}{l}^{\frac{1}{2}}\left(\frac{B_2 - B_1}{2\sigma}\right)^l h_{j-l}\left(\frac{\frac{B_1+B_2}{2} - \mu}{\sigma}\right)\int_{\mathbb{R}}\phi(t)f(t)h_j\left(\frac{t - \mu}{\sigma}\right) dt\right],$$

where $\phi(t) = \frac{1}{\sqrt{2\pi\sigma^2}} e^{-\frac{(t-\mu)^2}{2\sigma^2}}$, and the polynomials $h_n$ are defined in Equation 5.

*Proof.* First, we check $h'_n(x) = \sqrt{n} h_{n-1}(x)$ for $n \geq 1$, from

$$\frac{d}{dx} h_n(x) = \frac{(-1)^n}{\sqrt{n!}} \frac{d}{dx} \left[ e^{x^2/2} \frac{d^n}{dx^n} e^{-x^2/2} \right] = \frac{(-1)^n}{\sqrt{n!}} \left[ x e^{x^2/2} \frac{d^n}{dx^n} e^{-x^2/2} + e^{x^2/2} \frac{d^{n+1}}{dx^{n+1}} e^{-x^2/2} \right]$$

$$= \frac{(-1)^n e^{x^2/2}}{\sqrt{n!}} \left[ x \frac{d^n}{dx^n} e^{-x^2/2} + \frac{d^n}{dx^n} (-x e^{-x^2/2}) \right]$$

$$= \frac{(-1)^n e^{x^2/2}}{\sqrt{n!}} \left[ x \frac{d^n}{dx^n} e^{-x^2/2} + \left( -x \frac{d^n}{dx^n} e^{-x^2/2} + \binom{n}{1} \cdot (-1) \cdot \frac{d^{n-1}}{dx^{n-1}} e^{-x^2/2} \right) \right]$$

$$= \frac{(-1)^{n-1} n}{\sqrt{n!}} e^{x^2/2} \frac{d^{n-1}}{dx^{n-1}} e^{-x^2/2} = \sqrt{n} h_{n-1}(x).$$

Note that the coefficient $a_l$ can be obtained by $\frac{1}{l!} \frac{d^l}{dz^l} \Big|_{z=0} P_{\tilde{\mu}, \tilde{\sigma}^2}[d; \tilde{f}](z)$. From Theorem 2.2 and Lemma D.1,

$$a_l = \frac{1}{l!} \frac{d^l}{dz^l} \Big|_{z=0} P_{\tilde{\mu}, \tilde{\sigma}^2}[d; \tilde{f}](z) = \frac{1}{l!} \frac{d^l}{dz^l} \Big|_{z=0} P_{\mu, \sigma^2}[d; f] \left( \frac{B_2 - B_1}{2} z + \frac{B_2 + B_1}{2} \right)$$

$$= \frac{1}{l!} \frac{d^l}{dz^l} \Big|_{z=0} \sum_{j=0}^{d} h_j \left( \frac{\left( \frac{B_2 - B_1}{2} z + \frac{B_2 + B_1}{2} \right) - \mu}{\sigma} \right) \int_{\mathbb{R}} \phi(t) f(t) h_j \left( \frac{t - \mu}{\sigma} \right) dt$$

$$= \frac{1}{l!} \sum_{j=0}^{d} \left[ \frac{d^l}{dz^l} \Big|_{z=0} h_j \left( \frac{\left( \frac{B_2 - B_1}{2} z + \frac{B_2 + B_1}{2} \right) - \mu}{\sigma} \right) \right] \int_{\mathbb{R}} \phi(t) f(t) h_j \left( \frac{t - \mu}{\sigma} \right) dt$$

$$= \frac{1}{l!} \sum_{j=l}^{d} \left( \frac{B_2 - B_1}{2\sigma} \right)^l \left( \prod_{u=0}^{l-1} \sqrt{j - u} \right) h_{j-l} \left( \frac{\frac{B_1 + B_2}{2} - \mu}{\sigma} \right) \int_{\mathbb{R}} \phi(t) f(t) h_j \left( \frac{t - \mu}{\sigma} \right) dt$$

$$= \frac{1}{\sqrt{l!}} \sum_{j=l}^{d} \left[ \binom{j}{l}^{\frac{1}{2}} \left( \frac{B_2 - B_1}{2\sigma} \right)^l h_{j-l} \left( \frac{\frac{B_1 + B_2}{2} - \mu}{\sigma} \right) \int_{\mathbb{R}} \phi(t) f(t) h_j \left( \frac{t - \mu}{\sigma} \right) dt \right],$$

which completes the proof. $\qquad\square$

Using Lemma D.2, we now prove Theorem D.1 and D.2.

*Proof of Theorem D.1.* The proof directly follows from Lemma D.2, since

$$a_l = \left( \frac{B_2 - B_1}{2\sigma} \right)^l \left[ \frac{1}{\sqrt{l!}} \sum_{j=l}^{d} \binom{j}{l}^{\frac{1}{2}} h_{j-l} \left( \frac{\frac{B_1 + B_2}{2} - \mu}{\sigma} \right) \int_{\mathbb{R}} \phi(t) f(t) h_j \left( \frac{t - \mu}{\sigma} \right) dt \right],$$

and the term $\frac{1}{\sqrt{l!}} \sum_{j=l}^{d} \binom{j}{l}^{\frac{1}{2}} h_{j-l} \left( \frac{1}{\sigma} [\frac{B_1 + B_2}{2} - \mu] \right) \int_{\mathbb{R}} \phi(t) f(t) h_j \left( \frac{t-\mu}{\sigma} \right) dt$ is independent of the choice of $B_1$ and $B_2$ when $\frac{B_1 + B_2}{2}$ is fixed. $\qquad\square$

*Proof of Theorem D.2.* Considering $\lambda \tilde{\sigma} = (\lambda \sigma)/(\frac{B_2 - B_1}{2})$, the coefficient $a_l$ of the polynomial $\sum_{l=0}^{d} a_l z^l = P_{\tilde{\mu}, (\lambda \tilde{\sigma})^2}[d; \tilde{f}](z)$ is equal to

$$a_l = \frac{1}{\sqrt{l!}} \sum_{j=l}^{d} \left[ \binom{j}{l}^{\frac{1}{2}} \left( \frac{B_2 - B_1}{2\lambda\sigma} \right)^l h_{j-l} \left( \frac{\frac{B_1 + B_2}{2} - \mu}{\lambda\sigma} \right) \int_{\mathbb{R}} \phi_\lambda(t) f(t) h_j \left( \frac{t - \mu}{\lambda\sigma} \right) dt \right],$$

where $\phi_\lambda(t) = \frac{1}{\sqrt{2\pi\lambda^2\sigma^2}} e^{-\frac{(x-\mu)^2}{2\lambda^2\sigma^2}}$. Now, let

$$M := \sup_{j=l,l+1,\cdots,d,\ |u| \leq |\frac{1}{\sigma}(\frac{B_1+B_2}{2} - \mu)|} |h_{j-l}(u)|.$$

Note that $M < \infty$ is independent of the choice of $\lambda$. Also, from Theorem 2.2,

$$0 \le \|f - P_{\mu,(\lambda\sigma)^2}[d; f]\|_{\phi_\lambda} = \int_{\mathbb{R}} \phi_\lambda(t) f(t)^2 dt - \sum_{j=0}^{d} \left[ \int_{\mathbb{R}} \phi_\lambda(t) f(t) h_j \left( \frac{t-\mu}{\lambda\sigma} \right) dt \right]^2.$$

The above statement is valid since $\int_{\mathbb{R}} \phi_\lambda(t) f(t)^2 dt < \infty$, by assumption of $f$, dominated by linear function. Therefore,

$$\left| \int_{\mathbb{R}} \phi_\lambda(t) f(t) h_j \left( \frac{t-\mu}{\lambda\sigma} \right) dt \right|^2 \le \sum_{j=0}^{d} \left[ \int_{\mathbb{R}} \phi_\lambda(t) f(t) h_j \left( \frac{t-\mu}{\lambda\sigma} \right) dt \right]^2 \le \int_{\mathbb{R}} \phi_\lambda(t) f(t)^2 dt.$$

Using the assumption $|f(t)| < |\beta t + \gamma|$,

$$\int_{\mathbb{R}} \phi_\lambda(t) f(t)^2 dt = \int_{\mathbb{R}} \frac{1}{\sqrt{2\pi}} e^{-u^2/2} f(\mu + \lambda\sigma u)^2 du$$

$$< \int_{\mathbb{R}} \frac{1}{\sqrt{2\pi}} e^{-u^2/2} |\beta(\mu + \lambda\sigma u) + \gamma|^2 du = \beta_2 |\lambda|^2 + \beta_1 |\lambda| + \beta_0$$

for some $\beta_0$, $\beta_1$, $\beta_2 > 0$, independent of the choice of $\lambda$. As a result, for $l \ge 3$ and $\lambda > 1$,

$$|a_l| \le \frac{\beta_2 |\lambda|^2 + \beta_1 |\lambda| + \beta_0}{|\lambda|^l} \sum_{j=l}^{d} \left[ \binom{j}{l}^{\frac{1}{2}} \left( \frac{B_2 - B_1}{2\sigma} \right)^l \cdot M \right] \le \frac{C_l}{|\lambda|^{l-2}},$$

where $C_l = (\beta_2 + \beta_1 + \beta_0) \sum_{j=l}^{d} \left[ \binom{j}{l}^{\frac{1}{2}} \left( \frac{B_2 - B_1}{2\sigma} \right)^l \cdot M \right]$, which is also independent constant of the choice of $\lambda$. $\qquad \square$

