# OpenReview forum: "Optimizing Layerwise Polynomial Approximation for Efficient Private Inference on Fully Homomorphically Encryption: A Dynamic Programming Approach"
_ICLR.cc/2024/Conference — Submitted to ICLR 2024_

### Official Review · Reviewer_ov7g · 2023-10-27

**Soundness:** 3 good
**Presentation:** 2 fair
**Contribution:** 2 fair
**Rating:** 5
**Confidence:** 4

**Summary:**

This paper proposes a layer-wise approximation of activations to reduce the FHE-based secure inference time. By considering the characteristics of each layer, the proposed layer-wise approximation does not require re-training while maintaining the model performance (e.g., accuracy). In terms of polynomials approximations, this paper employs the weighted least squares approximation method with the input distribution to approximate the activation function and utilizes a dynamic programming algorithm to reduce the approximation polynomials’ degrees. In terms of the FHE ciphertext evaluation, the authors propose modulating the ciphertext moduli-chain layer-wise to reduce the inference time. Compared to prior works, this paper reduces the secure inference time by over 3 times on ResNet-20/ResNet-32 on dataset CIFAR-10 with negligible accuracy loss.

**Strengths:**

1. This paper studies the input distribution of the activation, uses the weighted least squares approximation method, and presents a dynamic programming algorithm designed to determine the optimal degree set for each layer. This algorithm efficiently obtains the optimal polynomial degrees, minimizing the overall negative impact on classification accuracy. The authors give a detailed mathematical analysis of the polynomial approximation method and dynamic programming design.
2. The authors propose a moduli-chain management method to achieve additional reductions in inference runtime. This method focuses on removing unused moduli. Instead of using a single moduli chain, the authors propose to use multiple moduli chains for each depth of the activation approximation polynomial function. In this way, this paper can reduce the number of bootstrapping operations.

**Weaknesses:**

1. The authors only evaluate the proposed method on ReLU. However, other activation functions, e.g., Swish, Sigmoid, GeLU, are not evaluated.

2. Some visualization of (1) the ReLU function, (2) the proposed approximation, and (3) other baselines will be helpful to understand the effect of the proposed approximation.

3. The authors need to show the improvement introduced by each technique, including the weighted MSE over minimax, layerwise over uniform, dynamic programming over simple greedy, etc. Right now, when compared with Lee et al., 2022a, it seems most of the benefits come from minimizing the weighted MSE, which drastically reduces the polynomial order.

4. Only a simple dataset, i.e., Cifar-10, is demonstrated. Larger datasets are important as they may impact the distribution of the activation functions as well as the required approximation accuracy.

**Questions:**

Please refer to the weaknesses.

---

> ### Author Response · Authors · 2023-11-21
> **Comments for Reviewer ov7g**
>
> Thank you for providing detailed insights into the aspects that need to be supplemented in our research. Your reviews are greatly appreciated.
>
> **Simulations for other activations**: The optimized approximation method we propose is adaptable to other activation functions, as outlined in Theorem 2.2. Given that this theorem establishes the minimization of mean squared error (MSE) by the derived polynomial, we can theoretically regard it as the optimal polynomial for minimizing inference errors. Furthermore, the ReLU function, in contrast to other activation functions, is non-differentiable, which can result in substantial errors.
>
> Also, our findings show that when approximating ReLU using the weighted least squares method, the MSE is higher than other differentiable activation functions (e.g., Swish, Sigmod, GELU). See the table below which presents the minimized for each activation where $\mu=0$ and $\sigma^2=1$. The minimized MSE can be evaluated by the second equation in Theorem 2.2.
>
> | Degree | ReLU | GELU | Swish | Sigmoid |
> |:---:|:---:|:---:|:---:|:---:|
> | 0 | 0.34085 | 0.34564 | 0.31308 | 0.04338 |
> | 1 | 0.09085 | 0.09564 | 0.06308 | 0.00069 |
> | 2 | 0.00612 | 0.00113 | 0.00154 | 0.00069 |
> | 3 | 0.00612 | 0.00113 | 0.00154 | 0.00004 |
>
> The efficacy of our method in approximating the challenging ReLU function supports that it should perform effectively in approximating other activation functions as well.
>
> **Visualization for the proposed approximate polynomial**: In the final version of the paper, we will include a graph comparing the characteristics of the existing minimax polynomial with those of the proposed polynomial. This will aid in better understanding the differences between the two.
>
> **Contributions for each proposed technique**: We provide the simulation results that show the improvement by each technique.
>
> First, we compare the results of the minimax polynomial and weighted LS with uniform degree for each layer. We additionally simulate on ResNet pre-trained with CIFAR-10. The minimax polynomial results are presented in Table 2. Applying the minimax polynomial with a degree of 2,835 (depth=13) leads to a degradation in the performance of the original model by -1.92%. However, the weighted LS method does not require such a large degree (or depth) to maintain the performance of the original model. See the table below providing the results of polynomial approximation with weighted LS.
>
> | Backbone | Original | Degree=15(Depth=4) | 31(5) | 63(6) | 127(7) | 255(8) |
> |:---:|:---:|:---:|:---:|:---:|:---:|:---:|
> | ResNet-20 | 91.52% | 88.50% | 91.13% | 91.70% | 91.60% | 91.59% |
> | ResNet-32 | 92.49% | 86.02% | 91.24% | 92.39% | 92.66% | 92.62% |
>
> Our simulation indicates that approximate polynomials generated by the weighted LS method with degrees 31 and 63 are sufficient to maintain the original classification accuracy for ResNet-20 and 32, respectively.
>
> Next, we compare the method of using different polynomial degrees per layer with the approach of using a uniform degree. The table below compares the expected time latency ($\sum_{i=1}^L T_{i}^{\text{Rel},\nu}(d_i)$, see equation 4 in the paper) required by each degree-setting method at the point where there is a 1% degradation in the performance of the original model .
>
> | Backbone | Uniform | Greedy | Proposed |
> |:---:|:---:|:---:|:---:|
> | ResNet-20 | 347.50 | 340.00 | 339.50 |
> | ResNet-32 | 652.50 | 600.75 | 600.25 |
>
> As evident from the table above, when applying a uniform degree, the time latency is highest, while utilizing dynamic programming to employ an optimal depth results in the lowest latency. Hence, our approach can be considered the most efficient in terms of minimizing time latency.
>
> **Simulation for various dataset**: We agree that the various simulations for many datasets are important. However, the challenges for large dataset in FHE-based PI still need further exploration. Pre-trained models that classify such datasets tend to have more intricate functions requiring optimization. For instance, non-linear functions such as max-pooling involve complex multivariate functions, and there hasn't been a suggested method to effectively approximate these functions in encrypted data for use in DNN inference. Moreover, due to current time latency issues, attempts to classify extensive datasets like TinyImageNet without retraining are restricted to operating on the final layers with encrypted data using FHE [1].
>
> [1] D. Kim et al, "Optimized Privacy-Preserving CNN Inference With Fully Homomorphic Encryption." IEEE Transactions on Information Forensics and Security 18 (2023): 2175-2187.

---

> > ### Comment · Reviewer_ov7g · 2023-12-04
> > **Reply for the author**
> >
> > Thanks for the reply.
> >
> > 1) Evaluation on a single dataset, i.e., Cifar10, is insufficient. The authors should at least have results on Cifar100. [1] actually demonstrates results on Imagenet.
> >
> > 2) The latency comparison between the proposed dynamic programming approach and the greedy approach is very close. The significance of the proposed approach is well demonstrated. Also, the accuracy difference between the greedy and the proposed approach needs to be shown.
> >
> > Hence, I will keep my original scores.
> >
> > [1] Moran Baruch et al., Training Large Scale Polynomial CNNs for E2E Inference over Homomorphic Encryption

---

### Official Review · Reviewer_8y16 · 2023-10-29

**Soundness:** 2 fair
**Presentation:** 3 good
**Contribution:** 2 fair
**Rating:** 5
**Confidence:** 3

**Summary:**

1. The paper proposes a layerwise degree optimization method for activation functions in fully homomorphic encryption (FHE) to reduce inference time while maintaining classification accuracy.
2. The previous approaches approximated activation functions uniformly, but this work takes into account the unique characteristics of each layer.
3. The simulations were performed using the Lattigo library on a high-performance computing system.

**Strengths:**

1. The author target the polynomial approximation problem for private Inference acceleration, which is a necessary and essential part for FHE-based private inference.
2. The authors employ the weighted least squares approximation method and optimize the degrees of activation functions using a dynamic programming algorithm.
3. They also propose modulating the ciphertext moduli-chain layerwise to further reduce inference time.
4.Experimental results on the CIFAR-10 dataset using the ResNet model show that the proposed method significantly reduces inference times compared to previous approaches.

**Weaknesses:**

1. I just have one major concern. There exist some work use 2-degree polynomial approximation [1,2] with ignorable accuracy loss. How does the proposed method compared to other advanced low-degree polynomial approximation techniques? As a low-degree polynomial approximation would outperforms the results in this work.

Reference:

[1] Park, Jaiyoung, et al. "AESPA: Accuracy preserving low-degree polynomial activation for fast private inference." arXiv preprint arXiv:2201.06699 (2022).
[2] Peng, Hongwu, et al. "PASNet: Polynomial Architecture Search Framework for Two-party Computation-based Secure Neural Network Deployment." 2023 60th ACM/IEEE Design Automation Conference (DAC). IEEE, 2023.

**Questions:**

See Weakness part.

---

> ### Author Response · Authors · 2023-11-21
> **Comments for Reviewer 8y16**
>
> Thank you for recognizing that the polynomial approximation problem is a crucial aspect in reducing latency for private inference (PI).
>
> Many studies employ low-degree polynomials as activation functions in PI using multi-party computation (MPC) protocols, allowing data sharing among multiple parties. Both of the studies you presented are MPC-based works which achieved high performance using low-degree polynomials. In contrast, our work focuses solely on encrypted data inferences using fully homomorphic encryption (FHE). FHE-based PI enables a server to perform homomorphic computations by itself without requiring additional communication costs.
>
> Within the realm of PI, MPC-based methodologies, similar to our study, tackle similar challenges and hold significance owing to unique advantages, like precise computation of non-arithmetic functions such as ReLU. Yet, directly contrasting these fields presents hurdles. Each prioritizes different optimization facets: FHE-based PI focuses on minimizing computation time for overall inference, whereas MPC-based PI grapples with substantial impediments in communication costs during computations. Despite recent efforts in MPC-based approaches employing low-degree polynomials for activation functions, their drawback lies in the substantial consumption of several gigabytes of communication costs [1]. Previous studies lack comprehensive quantitative assessments between FHE-based and MPC-based PI. Depending on the context, MPC-based methodologies might prove beneficial, while FHE-based approaches could offer distinctive advantages. Acknowledging the significance of both technologies is pivotal, given their representation as critical domains in continual progression.
>
> [1] P. Mishra et al, "Delphi: A Cryptographic Inference System for Neural Networks," in Proc. Workshop Privacy-Preserving Mach. Learn. Pract., Nov. 2020.

---

### Official Review · Reviewer_ghjk · 2023-10-31

**Soundness:** 2 fair
**Presentation:** 2 fair
**Contribution:** 2 fair
**Rating:** 3
**Confidence:** 4

**Summary:**

This paper presents a novel approach to enable private inference by using only Homomorphic Encryption (HE), without the need for retraining or model redesign.  The authors achieve this by replacing ReLUs with polynomial functions. They employ dynamic programming techniques and leverage layer-specific characteristics to adaptively select polynomial degrees for different layers in a pre-trained model.

**Strengths:**

1.  Authors exploited the layer-specific characteristics to significantly reduce the degree of polynomials for HE-only PI.

2. The proposed approach does not require any redesigning or fine-tuning of the model, which often helps recover the accuracy.


3. Methods are very well presented in the paper.

**Weaknesses:**

$\bullet$ **Complexity and scalability of the proposed approach:** An essential concern regarding the presented solution lies in its complexity, an aspect left unexplored in the paper. Given that the method determines the appropriate polynomial degree for each layer, its computational complexity varies depending on the network's depth. This adaptability makes it increasingly impractical for deeper networks, such as ResNet101 and ResNet152. In contrast, the complexity of PI-specific manual ReLU pruning depends solely on the number of stages within the networks and remains independent of the total number of layers.

Additionally, the authors must have included a comparative analysis of their approach's complexity with previous methods that involve retraining, with a focus on absolute time, to demonstrate the advantages of their method, which eliminates the need for retraining.

$\bullet$ **Lack of comprehensive empirical evaluation:**  The experimental evaluation in the paper is limited to CIFAR-10 using ResNet20 and ResNet32 networks. As per the standard practice in PI [1, 2, 3], an evaluation on CIFAR-100 and TinyImageNet datasets with networks such as ResNet18 should have been included to validate the effectiveness of the proposed solution on complex datasets.


$\bullet$ **Comparison with prior work:** To demonstrate the superiority of the HE-only solution over a hybrid approach (HE + MPC) [1, 2, 3],  a comparative analysis of the end-to-end PI runtime is required. Also, a discussion on low-degree polynomial substitution should have been included, which has been shown to be effective even on complex datasets such as [4].


**In conclusion,** I question the practicality of using higher-degree polynomials, especially considering their need for extensive bootstrapping when compared to the hybrid approach (HE/VOLE for linear layers and GC/OT for ReLUs). Additionally, this paper doesn't offer any new insights to enhance the feasibility of PI.

1. Kundu et al., “Learning to Linearize Deep Neural Networks for Secure and Efficient Private Inference,” ICLR 2023.

2. Cho et al., “Selective network linearization for efficient private inference,” ICML 2022.

 3. Jha et al., “DeepReduce: ReLU reduction for fast private inference,” ICML 2021.

4.  Chrysos et al., "Regularization of polynomial networks for image recognition," CVPR 2023

**Questions:**

Given that Table 1 indicates the need for a high-degree polynomial in the initial layer, is it possible to solely utilize the identity connection in these layers? Previous research on PI-specific ReLU optimization has shown that the initial ReLUs are not critical and can be eliminated without significant performance degradation. Is this applicable here?

---

> ### Author Response · Authors · 2023-11-21
> **Comments for Reviewer ghjk -1-**
>
> Thank you for valuable comments and understanding that our method can be applied in situations where redesigning and fine-tuning are not allowed.
>
> **Complexity and scalability of the proposed approach**: We agree that the computational complexity of our method depends on the network’s depth. In Section 3.2, page 7, we provide the complexity of the proposed dynamic programming algorithm with big O notation.
> However, once we obtain the set of the polynomial degrees from the proposed algorithm, we can perform the inference on all test datasets without additional computations for determining polynomials. Furthermore, the operations in the proposed algorithm involve operations between plaintexts, not ciphertexts. Therefore, we can efficiently obtain the optimal set of the polynomial degrees within a short timeframe. Consequently, obtaining the optimal polynomials for a given backbone network is expected to pose no significant burden when performing inference on the dataset.
>
> **Comparative analysis with works involving retraining**: We firstly want to note that the suggested works [1-3] are based on MPC protocol, and [4] requires retraining for achieving low degree polynomial activations.
>
> When considering the computational complexity of two methods, focusing solely on inference time is unfair. This is because one must also account for the time required to retrain the model. When retraining is involved, the methods to retrain can vary widely based on the size of the target dataset or the structure of the backbone model, making it challenging to precisely estimate the retraining time. As the target dataset or backbone model grows in complexity, the time needed for retraining is likely to increase. However, if effective retraining allows for achieving high performance using lower-degree polynomials, the resulting inference time may decrease.
>
> In scenarios like our research, where retraining is not allowed and the given backbone model must be used as-is, the time complexity will encompass finding the optimal polynomial approximation and the inference time. For our proposed algorithm, as stated in Section 3.2, the time complexity is denoted by the number of layers $N_L$ that need to be approximated, remaining independent of the model's structure or dataset size. Thus, the time spent finding the optimal polynomial is considerably shorter compared to the time required for retraining. Of course, in contrast to cases where retraining is allowed, the degree of the approximating polynomial demanded is higher, resulting in relatively longer inference times.
>
> In summary, the consideration of time complexity in PI involves two different scenarios:
> - Involving retraining: Retraining time + Inference time
> - Not allowing retraining: Polynomial optimizing time + Inference time
>
> Both approaches are crucial in PI research and can be applicable depending on the circumstances. If the model owner providing PI services has sufficient resources to train the model and enough time for retraining, applying the polynomial approximation technique allowing retraining might be preferable. However, situations might arise where resources or time for retraining are insufficient, and there's no guarantee that retraining always yields optimal results. In such cases, performing rapid polynomial approximation tailored to the already trained model, as proposed in our research, could be more favorable. Therefore, directly comparing the time consumption between PI with and without retraining is challenging, and previous studies have not extensively explored this direct comparison.
> Our research aimed to reduce the bottleneck of the polynomial degree, which becomes the time complexity issue in scenarios where retraining is not allowed. By minimizing this aspect more than in previous studies, we believe our work contributes to reducing the overall time spent in PI.
>
> Such distinctions in research aren't exclusive to PI. For instance, the domain of quantizing DNNs, where researchers also differentiate between areas that allow retraining and those that don't. Some studies [5] aim for high model performance by permitting retraining in the pursuit of efficiently executing inference through quantization. Conversely, other studies [6] focus on finding appropriate precision bits without retraining to minimize performance degradation. Similar to the earlier discussion, in both these domains, the objectives and optimization targets vary, making it challenging to definitively label one as superior to the other. Considering that research in PI can progress independently based on the allowance of retraining, it would be valuable to acknowledge these aspects.
>
> [1-4] The papers that mentioned in the review.
>
> [5] S. Anwar et al, "Fixed Point Optimization of Deep Convolutional Neural Networks for Object Recognition," ICASSP, IEEE, 2015.
>
> [6] D. Lin et al, "Fixed Point Quantization of Deep Convolutional Networks," ICML, 2016.

---

> ### Author Response · Authors · 2023-11-21
> **Comments for Reviewer ghjk -2-**
>
> **Comparison between previous works with hybrid approach and our work**: Both approaches, the hybrid approach (allowing MPC) and the FHE-only approach, share the common goal of performing privacy-preserving inference. However, directly comparing these domains is impracticable due to differences in their focus areas: these approaches deal with *distinct domains* and optimize *different bottlenecks*.
>
> *Distinct domain.* We kindly ask for consideration regarding the fact that research in these two domains: hybrid and FHE-only, is conducted for distinct environmental aspects. If the client can participate in computations alongside the server during PI service provision, employing a hybrid approach is desirable. However, situations may arise, due to communication constraints (such as the client being intermittently offline) or for security reasons concerning the server's computational process, where client involvement during inference is not permitted. In such cases, the hybrid model becomes unfeasible, necessitating the exclusive use of an FHE-only model. Therefore, the enhancement of FHE-only models a crucial research focus on some scenarios. This is why our current line of work has been shaped: to improve upon FHE-only models by comparing and contrasting them with their counterparts.
>
> *Different bottleneck.* Each approach prioritizes different optimization aspects: FHE-only PI focuses on minimizing computation time for overall inference, while hybrid PI addresses significant challenges related to communication costs during computations. Despite recent MPC-based attempts to use low-degree polynomials for activation functions, their drawback lies in their consumption of several gigabytes of communication cost [7]. Due to these inherent and critical distinctions, quantitatively comparing these challenges is challenging. To our knowledge, there lacks proper papers that fairly compare FHE-based PI and MPC-based PI.
>
> Moreover, hardware accelerators have effectively addressed the inference time, previously a bottleneck in FHE-only approaches. Several studies [8-10] demonstrate the practicality of executing homomorphic operations and inference on DNNs using the RNS-CKKS scheme with FHE’ed data. Our research compares the effects of reducing the degree of the approximating polynomial in situations where the backbone network mandates the direct utilization of given pre-trained parameters without retraining, contrasting this with conventional minimax approximations [11,12]. Directly comparing the overall end-to-end PI runtime with other hybrid works seems difficult. However, with the ongoing advancements in hardware accelerators and related technologies, integrating them with our layerwise optimization gives ample reason to anticipate a further reduction in the overall runtime.
>
> Summing up, the PI researches on hybrid approach and FHE only approach exhibit distinct differences, each serving specific environments and application domains. Consequently, independent research is imperative in both areas to accommodate their unique applicability.
>
> [7] P. Mishra et al, "Delphi: A Cryptographic Inference System for Neural Networks," in Proc. Workshop Privacy-Preserving Mach. Learn. Pract., Nov. 2020.
>
> [8] S. Kim et al, "BTS: An Accelerator for Bootstrappable Fully Homomorphic Encryption," Proceedings of the 49th Annual International Symposium on Computer Architecture. 2022.
>
> [9] SU. Yang, et al, "FPGA-based Hardware Accelerator for Leveled Ring-LWE Fully Homomorphic Encryption," IEEE Access, 2020.
>
> [10] H. Lee et al, "A 2.7~13.3 uJ/boot/slot Flexible RNS-CKKS Processor in 28nm CMOS Technology for FHE-based Privacy-Preserving
> Computing," IEEE ISSCC, San Francisco, CA, USA, Feb. 2024, accepted for presentation.
>
> [11] E. Lee et al, "Low-Complexity Deep Convolutional Neural Networks on Fully Homomorphic Encryption Using Multiplexed
> Convolutions," ICML, 2022.
>
> [12] D. Kim et al, "Optimized Privacy-Preserving CNN Inference With Fully Homomorphic Encryption." IEEE Transactions on Information Forensics and Security 18 (2023): 2175-2187, 2023.

---

> > ### Comment · Reviewer_ghjk · 2023-11-22
> > **Reply to Authors' Rebuttal**
> >
> > Thank you for a detailed rebuttal.
> >
> >
> > Here are a few notes
> >
> > $\bullet$ Auhtors' argument for the re-training time in the previous work, which used the HE and MPC, is not justified. Firstly, the re-training process in the these previous work, similar to the method proposed in this study, occurs in plaintext rather than ciphertext. Additionally, not all previous studies necessitated re-training. While fine-grained ReLU optimization [1,2] require re-training, while [3] does not, instead utilizing KD (applying KD is different than re-training, you need to just modify the loss function). Finally, in the context of private inference, the time investment made by a service provider in training a model is less critical. Once the model is deployed, the focus shifts to the efficiency and cost-effectiveness of the private inference.
> >
> > $\bullet$ On the last question
> >
> > Prior work [1-3] consistently demonstrated that, ReLUs in the initial layers (precisely,  Stage 1) can be dropped without impacting accuracy. In fact, [2] has the theoretical proof for the same. This observation is consistent across different network and datasets. Leverging this insights, authors could have reduced the requirement of higher degree polynomials in the initial layer of the ResNets.
> >
> > I still have do not see the practicality of using such a higher degree of polynomials, especially, when [5] demonstrated that HE-only solution can be achieved with **only degree 4 polynomials**, also scalable to ImageNet datasets.
> >
> > [5]  Baruch et al., Training Large Scale Polynomial CNNs for E2E Inference over Homomorphic Encryption.
> >
> >
> >  I will keep my score, given the current version of the paper.

---

> > > ### Author Response · Authors · 2023-11-23
> > > **Reply for the review**
> > >
> > > As mentioned in our previous comment, we believe our research stands out by being applicable in situations where retraining is not required. While conducting retraining in plaintext might seem less laborious in certain scenarios, it could become impractical, especially in the case of recent advancements such as large-scale language models (LLMs) that demand extensive resources for training.
> > >
> > > The field of research in deep neural network quantization also divides into two categories similar to the PI research: quantization-aware training (QAT), which awares retraining, and post-training quantization (PTQ), where quantization is performed without additional retraining on pre-trained models [6-8]. In the realm of PI, the hybrid approach utilizes retraining to facilitate inference with relatively low-degree polynomials, distinguishing itself from the FHE-only method. Similarly, in quantization, QAT permits inference with lower quantization bit precision compared to PTQ. However, there inevitably exists a significant trade-off between saving bit precision and training cost, and both research areas have continued to evolve independently. Moreover, there's a growing necessity for applications, like LLMs, where retraining becomes infeasible [8], prompting efforts to lightweight LLMs without retraining to minimize the degradation of model performance. As a result, the line of work enhancing performance in PTQ has become crucial.
> > >
> > > Please also recognize the utility of our retraining-free PI technology, similar to the significance observed in quantization research. In our paper, the FHE-only approach uses high-degree polynomials to approximate non-linear functions precisely, and thus PI can be performed without retraining. On the other hand, the MPC-FHE hybrid method using low-degree polynomials mentioned by the reviewer [1-5] requires retraining or training from scratch. Also, numerous studies have proven practicality of FHE-only approach, including validation through methods such as hardware acceleration [9-12]. Considering our technology's potential in reducing inference time latency for large models that do not allow retraining, we hope it is duly acknowledged.
> > >
> > > **For the last question,** we'd like to add two comments:
> > >
> > > - First, to our understanding, it has been theoretically demonstrated that dropping ReLU is applicable only to networks with three layers.
> > >
> > > - Second, theoretical properties outlined in [2] assume modifiability in weight parameters. The term ‘memorization capacity’ defined therein refers to the maximum number $N$ ensuring the existence of weight parameters $W$, satisfying $f_W(x_i)=y_i$ for a set of random $N$ choices $(x_i,y_i)$. However, our study focuses on scenarios where $W$ remains fixed (while noting that the model function can be altered by approximating activations, despite fixed $W$). In this context, we believe that the theorems presented in [2] need adjustments for application.
> > >
> > > [1-5] The papers mentioned by the reviewer earlier.
> > >
> > > [6] Liu, Zhenhua, et al. "Post-training quantization for vision transformer," NeurIPS 2021.
> > >
> > > [7] Yao, Zhewei, et al. "Zeroquant: Efficient and affordable post-training quantization for large-scale transformers," NeurIPS 2022.
> > >
> > > [8] Xiao, Guangxuan, et al. "Smoothquant: Accurate and efficient post-training quantization for large language models," ICML 2023.
> > >
> > > [9] N. Samardzic et al, "F1: A Fast and Programmable Accelerator for Fully Homomorphic encryption," MICRO-54: 54th Annual IEEE/ACM International Symposium on Microarchitecture, 2021.
> > >
> > > [10] S. Kim et al, "BTS: An Accelerator for Bootstrappable Fully Homomorphic Encryption," Proceedings of the 49th Annual International Symposium on Computer Architecture. 2022.
> > >
> > > [11] SU. Yang, et al, "FPGA-based Hardware Accelerator for Leveled Ring-LWE Fully Homomorphic Encryption," IEEE Access, 2020.
> > >
> > > [12] H. Lee et al, "A 2.7~13.3 uJ/boot/slot Flexible RNS-CKKS Processor in 28nm CMOS Technology for FHE-based Privacy-Preserving Computing," IEEE ISSCC, San Francisco, CA, USA, Feb. 2024, accepted for presentation.

---

> ### Author Response · Authors · 2023-11-21
> **Comments for Reviewer ghjk -3-**
>
> **Simulation for complex dataset**: Experiments could be conducted on the CIFAR-100 dataset to further demonstrate the effectiveness of our algorithm. We check the performance of our optimizing method by classifying plaintext CIFAR-100 on ResNet-32.
>
> | Method | Max. Degree | Max. Depth | Original Acc. | Approx. Acc. |
> |:---:|:---:|:---:|:---:|:---:|
> | Previous [11] | 6,075 | 14 | 69.50% | 69.43% |
> | Proposed | 511 | 9 | 69.49% | 68.94%* |
>
> (*The accuracy of the model using the proposed approximated polynomials is obtained in plaintext.)
>
> Our technique significantly reduced depth consumption while maintaining the performance of the pre-trained model. Since the maximum depth consumption is reduced from 14 to 9, we can anticipate that the computation time for bootstrapping and ReLU evaluation will also be significantly reduced in ciphertext CIFAR-100 classification using the proposed method. Given the limited discussion timeframe, demonstrating other outcomes such as ciphertext experiments might be challenging at present. However, efforts will be made to sufficiently supplement the experimental content before the final revision deadline.
>
> FHE-based PI still presents several unresolved issues necessitating further exploration, prompting us to propose future work for classifying larger datasets. Pre-trained models capable of classifying large datasets often contain more complex functions requiring optimization. For instance, non-arithmetic functions like max-pooling involve multivariate functions, and there hasn't been a proposed approach to effectively approximate these in encrypted data for inference in DNNs. Additionally, due to the current time latency issues, instances of classifying large datasets like TinyImageNet without retraining are limited to performing on the last few layers with FHE'ed data [12].
>
> **Reply for the last question**: We note that the optimized polynomial degrees would depend on the datasets, the trained models, and the retraining process. While not directly within the scope of PI research, analogous studies in the field of quantization–specifically, those focusing on quantizing weights and activations without retraining the model–have indicated the need for higher bit-precision in the initial layers during quantization [6]. This observation is consistent with our research, where we use higher-degree polynomials to approximate the initial ReLU functions. Additionally, if our method were to be employed in scenarios permitting retraining as in prior MPC-based work, the initial ReLU functions might be eliminated.
>
> [11] E. Lee et al, "Low-Complexity Deep Convolutional Neural Networks on Fully Homomorphic Encryption Using Multiplexed Convolutions," ICML, 2022.
>
> [12] D. Kim et al, "Optimized Privacy-Preserving CNN Inference With Fully Homomorphic Encryption." IEEE Transactions on Information Forensics and Security 18 (2023): 2175-2187, 2023.

---

### Official Review · Reviewer_J1kw · 2023-11-05

**Soundness:** 2 fair
**Presentation:** 3 good
**Contribution:** 2 fair
**Rating:** 5
**Confidence:** 4

**Summary:**

The paper makes two main contributions to reduce the inference latency of deep convolutional neural networks (CNNs such as ResNet-20/32) when the input data is encrypted using a fully homomorphic encryption (FHE) scheme (RNS-CKKS).

1) Assuming that the input distribution to the ReLU activation function is a normal distribution, it has been claimed that optimizing the mean squared error (MSE) of the polynomial approximation is better than the conventional minimax approach (which assumes uniform distribution). This has been achieved by tying it to minimizing the variance of the loss function.

2) A dynamic-programming based method has been proposed to determine "optimal" polynomial degree in each layer of a neural network. Based on this optimization approach, it have been shown that the inference latency can be reduced 3-4x compared to the baseline.

**Strengths:**

The paper address an important problem in the field of private inference and the proposed solution appears to be based on solid principles, provided that the stated assumptions are true.

**Weaknesses:**

1) Two key claims in the paper have been stated without any strong validation.

a) Firstly, what is the guarantee that the input to the ReLU activation layer will always follow a normal distribution? Is this true for every layer of the neural network? What happens if this assumption does not hold?

b) More importantly, why is the "variance of the loss" a good surrogate for the classification accuracy? The challenge in encrypted domain inference is not sensitivity to small approximation errors. Contrarily, even a single large (unbounded) approximation error can screw up the entire inference process (as observed in the bit-flip attack on machine learning models). This is reason we need some bound on the approximation error (leading to the minimax formulation).

2) It is well-established in the literature on MPC-based private inference that not all ReLUs are equally important in the inference process. For example, see Peng et al., "AutoReP: Automatic ReLU Replacement for Fast Private Network Inference", ICCV 2023 and the references therin. In fact, the literature on MPC-based private inference does not stop layer-wise and tries to find exactly which particular neuron requires more accurate approximation.

a) It is important to acknowledge the progress made on ReLU reduction in the field of MPC-based private inference because it is directly relevant to the problem considered in this work.

b) Can the proposed dynamic programming approach be scaled one step further to find the optimal polynomial degree for each specific neuron?

3) The results indicate that there is orders of magnitude decrease in the max. polynomial degree (see Table 3), which is surprising not reflected in the max. depth as well as the overall inference time. There should be a more in-depth analysis of how the max. polynomial degree impacts the inference time because that forms the core motivation for this work.

**Questions:**

Please see weaknesses.

---

> ### Author Response · Authors · 2023-11-21
> **Comments for Reviewer J1kw -1-**
>
> Thank you for your valuable comments and feedback that we need to take into account.
>
> **Assumption of normality**: In our simulations, we conducted empirical verification to confirm that the input values of each activation function follow a normal distribution by drawing a histogram of input data. We will include our empirical results supporting the normality of input distribution. It is also noteworthy that similar normality assumptions have been employed in prior studies [1, 2].
> Furthermore, our proposed method can be applied to arbitrary input distributions. Theorem 2.1 shows that we can obtain polynomials minimizing the mean squared error for arbitrary input distributions. We are grateful for your important comments. We will ensure our revised paper includes the initial assumption of normality and the general applicability of our work.
>
> [1] I, Takumi et al, "Highly accurate CNN Inference Using Approximate Activation Functions Over Homomorphic Encryption," 2020 IEEE International Conference on Big Data (Big Data). IEEE, 2020.
>
> [2] P, Hongwu et al. "AutoReP: Automatic ReLU Replacement for Fast Private Network Inference," ICCV, 2023.
>
> **Discussion of the proposed variance of the loss**: We provide an intuition to explain why the variance of the loss increment $Var(\Delta L)$ is a proper surrogate in our optimization. Because of $\Delta \mathcal{L}= \frac{\partial \mathcal{L}}{\partial a_{i,j}} \Delta a_{i,j}$, where $a_{i,j}$ is a random variable representing a polynomial approximation error (see page 5), we can set $E[\Delta \mathcal{L}]=\frac{\partial \mathcal{L}}{\partial a_{i,j}} E[\Delta a_{i,j}]$. Our empirical findings indicate that $E[\Delta a_{i,j}]\approx 0$, which leads to $E[\Delta \mathcal{L}]\approx 0$. Hence, we can claim that $Var(\Delta L) \approx E[(\Delta L)^2]$.
> The practical implication of this approximation is important. By focusing on minimizing $Var(\Delta L)$, we can effectively reduce the noise power on the loss function (i.e., maximizing the signal to noise ratio (SNR) of loss).
>
> As you mentioned, we fully agree that a single large approximation error can severely compromise the total inference accuracy. This holds true even in the context of minimax approximation. Recognizing the impact of a single value on the overall inference, the configuration of the approximation range becomes crucial. Consequently, both prior research employing minimax approximation and our own study have considered the task of defining the approximation range [3,4].
>
> In the study employing minimax approximation, the approximation range should be set such that all input values of each activation function fall within the approximation range. In our approach, employing the weighted least squares method, we aimed to reduce the size of regions where polynomial values diverge significantly. To achieve this, we scaled the obtained standard deviation $\sigma$ of the input distribution as an approximation weight, as detailed in the last part of Section 2.2, page 4.
>
> Once an appropriate approximation range is established, the inference error can be considered as determined by the small errors in each layer. These small errors in each layer are expected to have varying impacts on the overall inference result. Through the discussion in Section 3.1, we argue that the diverse contributions of each layer directly correlate with the variance of the loss increment $Var(\Delta L)$. This discussion gains significance from the assumption that input values exist within appropriate approximation ranges, ensuring the absence of single large approximation errors in each layer.
>
> Our simulation results further support the validity of our approximation range settings. The minimal difference in classification accuracy before and after approximating the ReLU function with a polynomial supports our approach (see the results of the proposed method in Table 2).
>
> Another reason that the variance of the loss increment can be considered as a stable measure that indicates the classification accuracy is that the value $\Delta L$ is always positive. We can assume that the activation nodes are on the global minimum of the loss function. Then, the small errors of the activation function always result in positive change on the loss function. Therefore, the reduction of the variance $\Delta L$ leads the activation function to the global minimum, which gives better results in classification accuracy.
>
> [3] J.-W. Lee et al, "Privacy-Preserving Machine Learning with Fully Homomorphic Encryption for Deep Neural Network," IEEE Access, 10:30039–30054, 2022b.
>
> [4] E. Lee et al, "Low-Complexity Deep Convolutional Neural Networks on Fully Homomorphic Encryption Using Multiplexed Convolutions," ICML, 2022.

---

> ### Author Response · Authors · 2023-11-21
> **Comments for Reviewer J1kw -2-**
>
> **Comments about AutoReP**: Thank you for bringing the recent AutoReP study to our attention. We will acknowledge AutoReP and other pertinent research on MPC-based private inference in our paper. It's noteworthy that AutoReP introduces a method for selecting specific ReLU functions to replace with identical degree polynomials, which is technically distinct from our approach. In our work, due to the limitations of fully homomorphic encryption (FHE) in evaluating ReLU operations, we substitute all ReLU functions in nodes with fine-grained polynomials, which are optimized by dynamic programming approach.
> Moreover, AutoReP employs retraining to identify ReLU activation nodes for polynomial replacement. In contrast, our method does not require retraining, making it more suited for scenarios where retraining is not feasible.
> Additionally, we would like to note that both AutoReP and our work are considered contemporaneous according to the ICLR reviewer guidelines (https://iclr.cc/Conferences/2024/ReviewerGuide) because AutoReP was posted on arXiv in August 2023, a month prior to our submission in September 2023.
>
> **Optimizing for each specific neuron**: In AutoReP, a more optimized approximation approach is adopted by considering the characteristics of each neuron individually, rather than uniformly applying the same polynomial across the entire layer. While the RNS-CKKS scheme also allows applying different polynomials to each neuron within a single ciphertext, the RNS-CKKS scheme used in our FHE-based technology follows a single instruction multiple data (SIMD) approach, where multiple data are encoded into a single ciphertext for homomorphic operations. Consequently, even if different polynomials are applied to the values within a single ciphertext, the depth consumption of the ciphertext depends on the maximum degree among those polynomials. In FHE-based operations, the most crucial factor influencing computation time is the depth consumption of ciphertexts. Therefore, applying different polynomials to each neuron may not result in significant differences in overall computation time.
>
> However, our proposed dynamic programming technique can work independently of the SIMD structure, making it applicable to MPC-based schemes as well. By assessing the input distribution for each neuron and formulating an optimization problem with optimization variables for each degree (i.e., $d_{i,j}$ for $j$th node in the $i$th layer), we can optimize the polynomials tailored to each individual neuron.
>
> **The impact of degrees and depths on the inference time**: In FHE-based schemes, the total inference time is closely associated with the depth consumption of a polynomial rather than the degree. The depth consumption is almost proportional to the logarithm of the polynomial degree (depth consumption = $\lceil \log_2 (d+1) \rceil$, where $d$ is the degree of the polynomial).
>
> The analysis of the depth consumption and its relation to the computation time for the polynomials proposed in each layer is presented in Section 4. As illustrated in Figure 1, page 7, as the depth $\delta$ consumed by each polynomial in a layer decreases, both bootstrapping and polynomial evaluation times decrease (blue graph). Furthermore, experimental results using the moduli chain management method proposed in Section 4 confirm that as the depth of the approximated polynomials decreases, the computation time decreases even more (orange graph).
> By reducing depth consumption, we can effectively reduce the overall inference time. We provide detailed analysis on this aspect in Section 4.

---

### Meta-Review · Area_Chair_vNdj · 2023-12-04

**Metareview:**

The paper develops a dynamic programming algorithm for optimizing and significantly reducing the degree of polynomials for FHE. The proposed approach does not require any retraining of the model.

There seems a disagreement between the authors and the reviewers regarding the importance of considering/comparing to recent advances in the ReLU reduction in MPC-based private inference.  To avoid further arguments on this topic, perhaps the authors could include a comparative analysis of their approach's complexity with previous methods that involve retraining, in terms of absolute time, to demonstrate the advantages of their method more clearly.

Overall, I agree with the reviewers both on the merits of the proposed method and the concerns raised by them. Based on the consensus  of the reviewers' opinion, I recommend rejection.

**Justification For Why Not Higher Score:**

I agree with the reviewers regarding their concerns. See above.

**Justification For Why Not Lower Score:**

NA

---

### Decision · Program_Chairs · 2024-01-16

Reject